# Theory of optical axion electrodynamics and application to the Kerr effect in topological antiferromagnets

Junyeong Ahn [1] ✉, Su-Yang Xu [2] ✉ & Ashvin Vishwanath[1] ✉

Emergent axion electrodynamics in magneto-electric media is expected to provide novel ways to detect and control material properties with electromagnetic fields. However, despite being studied intensively for over a decade, its theoretical understanding remains mostly confined to the static limit. Here, we introduce a theory of axion electrodynamics at general frequencies. We define a proper optical axion magneto-electric coupling through its relation to optical surface Hall conductivity and provide ways to calculate it in lattice systems. By employing our formulas, we show that axion electrodynamics can lead to a significant Kerr effect in thin-film antiferromagnets at wavelengths that are seemingly too long to resolve the spatial modulation of magnetism. We identify the wavelength scale above which the Kerr effect is suppressed. Our theory is particularly relevant to materials like $MnBi_2Te_4$, a topological antiferromagnet whose magneto-electric response is shown here to be dominated by the axion contribution even at optical frequencies.

In a medium where spatial inversion and time reversal symmetries are both broken, magnetic and electric fields couple in a way that is fundamentally different from the electromagnetic induction described by Maxwell's equations in vacuum, thus leading to exotic electrodynamics. A topic of particular interest in magneto-electric coupling phenomena recently is axion electrodynamics. Axion electrodynamics is theoretically proposed to be realized in a class of topological materials called axion insulators[1–6] for sufficiently slowly oscillating electromagnetic fields. In particular, the static axion magneto-electric coupling is quantized by a multiple of the fundamental constant $e^2/2h$[7–9], which originates from the half-quantized Hall conductance of the topological surface states. While such a quantized magneto-electric coupling has not yet been directly observed, experimental progress has been made to observe its consequences in the static limit[10–16].

In contrast to the static limit, the emergent axion electrodynamics at generic optical frequencies remains largely unexplored theoretically. Formulating a theory of optical axion electrodynamics has to overcome the challenge that the axion coupling is ill-defined in periodic three-dimensional systems, making it hard to calculate and understand. The static axion angle can be calculated by the Chern-Simons integral of the non-abelian Berry connection[8,17,18] at the cost of being gauge dependent. Despite its gauge dependence, the Chern-Simons integral is well-defined as an angular variable taking a value between 0 and $2\pi$, because a gauge transformation changes its value only by a multiple of $2\pi$. This is a manifestation that the static surface Hall conductance changes by a multiple of $e^2/h$ under deformations. However, at optical frequencies, a similar approach does not seem feasible. This is because the optical surface Hall conductivity is frequency dependent, and thus a generic surface deformation changes the surface Hall response by a non-quantized amount. Owing to this difficulty, theoretical understanding of the optical axion electrodynamics has remained elusive to date[19].

In this paper, we make two important steps toward the complete formulation of the optical axion electrodynamics. First, we show that a proper definition of the optical magneto-electric coupling allows us to calculate the optical axion angle in a fully gauge-independent way in a system with finite thickness. Although the optical axion electrodynamics is a part of the linear-response optical magneto-electric effects, it is distinguished from the other contributions. Non-axionic magneto-electric effects are described within the well-

---

[1]Department of Physics, Harvard University, Cambridge, MA 02138, USA. [2]Department of Chemistry and Chemical Biology, Harvard University, Cambridge, MA 02138, USA. ✉e-mail: junyeongahn@fas.harvard.edu; suyangxu@fas.harvard.edu; avishwanath@g.harvard.edu

established theory of gyrotropic birefringence and natural optical activity, which are based on the bulk current response[19–21]. On the other hand, the optical axion electrodynamics is a surface phenomenon and thus is not captured by those theories[19]. We therefore define the well-define optical axion angle by analyzing the surface response. Using the well-defined axion angle, we can understand the optical axion electrodynamics of thin films and also that of bulk crystals by increasing the thickness. Second, in systems with periodic boundary conditions along all three directions, we find the optical axion angle at high optical frequencies can be estimated by the optical layer Hall conductivity that we define in the main text. These findings present advantages in numerical calculations as well as conceptual advances.

Our development of a theory of optical axion electrodynamics opens the door to understanding novel axion-induced optical phenomena. Here we provide a concrete example. Magneto-optic effects make electromagnetic waves powerful probes of the magnetic structure in materials. Conventionally, both Kerr and Faraday effects are attributed to the optical Hall effect and thus commonly perceived as probes of net magnetization[22,23]. Although not well known, previous theoretical studies proposed that the magneto-electric effect can also lead to the Kerr effect, providing a novel way to probe fully compensated antiferromagnets whose symmetries strictly prohibit the Hall effect and net magnetization[20,24–28]. Still, however, there are two aspects that need further investigation. First, axion electrodynamic contribution to the Kerr effect has not been well understood. Second, since previous studies have focused on three-dimensional bulk systems, magneto-optic Kerr effects in quasi two-dimensional antiferromagnets remain elusive. We present theoretical analysis revealing the precise conditions for realizing the optical axion electrodynamic Kerr effect as well as quantitative numerical analysis allowed by our gauge-invariant formulas.

Our results apply broadly to both topological and non-topological media because optical magneto-electric effects, as non-quantized phenomena, are not sensitive to the topological nature of the ground state. Therefore, our work provides a theoretical basis for the detection and manipulation of antiferromagnetism in a large class of materials, thus having potential for wide applications to antiferromagnetic spintronics and the study of magnetic structure in quantum materials

Meanwhile, topological antiferromagnets need special attention as they are ideal platforms for optical axion electrodynamics. To understand this, we note two aspects. First, it is desired to have antiferromagnets having bulk symmetry that reverses an odd number of spacetime coordinates and is broken on the surfaces (e.g., spatial inversion symmetry), because such a symmetry suppresses bulk magneto-electric effects but not the axion magneto-electric effect. This condition is similar to the requirement of the quantization of the static axion angle but additionally requires that the quantizing-symmetry is broken at the surface to allow for a nonzero value. Second, spatially spreading of electronic quantum states is needed, in order to show a response distinguished from that of decoupled layered or Mott antiferromagnets. This condition is again satisfied in topological antiferromagnets. We thus apply our theory to a model of MnBi$_2$Te$_4$, which is the only experimentally realized axion topological antiferromagnet to date, and show that the Kerr effect in this material is significant.

## Results

### Optical magneto-electric coupling

Motivated by the equivalence between the surface Hall conductivity and the axion angle in the static limit, we study the surface current response to define the optical axion angle. We consider currents generated by electromagnetic multipole moments. Electromagnetic responses from multipole moments are smaller for higher order

moments[21]: electric dipole $P_i \gg$ electric quadrupole $Q_{ij}$ and magnetic dipole $M_i \gg$ higher orders. Here, we consider only up to the electric quadrupole-magnetic dipole order, giving the leading-order magneto-electric effect. The bulk current density is related to the multipole moments by: $J_i = \dot{P}_i - \frac{1}{2}\partial_j \dot{Q}_{ij} + \epsilon_{ijk}\partial_j M_k$. While electric quadrupole and magnetic dipole moments do not generate macroscopic currents in macroscopically homogeneous lattice systems, they generate currents on the system boundary where the material property changes spatially. The induced multipole moments have the following form in the frequency domain[21]:

$$P_i = \sum_j (\chi_{ij} - i\chi'_{ij})E_j + \frac{1}{2}\sum_{jk}(a_{ijk} - ia'_{ijk})\nabla_k E_j + \sum_j (G_{ij} - iG'_{ij})B_j,$$

$$Q_{ij} = \sum_k (a_{kij} + ia'_{kij})E_k, \qquad (1)$$

$$M_i = \sum_j (G_{ji} + iG'_{ji})E_j,$$

where $\chi_{ij} = \chi_{ji}$ and $\chi'_{ij} = -\chi'_{ji}$ are the electric susceptibility tensors, $G_{ij}$ and $G'_{ij}$ are magneto-electric coupling, and $a'_{ijk} = a'_{ikj}$ and $a_{ijk} = a_{ikj}$ are electric quadrupolar susceptibility tensors. Here, we are interested in the magnetic magneto-electric effects described by $G_{ij}$ and $a'_{ijk}$, which occur only when time reversal symmetry is broken while being compatible with spacetime inversion $PT$ symmetry. Therefore, we assume $PT$ symmetry to neglect the complications arising from the bulk Hall conductivity and natural optical activity, both of which are excluded in this symmetry setting, i.e., $\chi'_{ij} = 0$ and $G'_{ij} = a_{ijk} = 0$ (Table 1). This assumption does not affect our key results (see Methods for discussions without $PT$ symmetry). As we show below, magneto-electric effects occur in combination with electric quadrupole responses at optical frequencies, requiring the consideration of the combination of $G_{ij}$ and $a'_{ijk}$.

Let us consider the surface at $z = 0$ of a three-dimensional homogeneous material with the outward normal direction $\hat{z}$ (Fig. 1). The surface current density $\mathbf{j}^s = \int_{-d_s/2}^{d_s/2} dz \mathbf{J}$, where $d_s$ is the characteristic thickness of the interface where response functions change rapidly as functions of $z$, is related to the multipole moments through

$$j_i^s(\omega) = \sum_{j,k}\left[\delta G_{jk}(\omega) + \sum_l \frac{\omega}{2}\epsilon_{kzl}\delta a'_{jlz}(\omega)\right]\epsilon_{kzi}E_j(\omega)$$
$$+ \sum_j \left[\bar{\sigma}_{ij}(\omega) - i\bar{\sigma}_{izj}(\omega)\right]\delta E_j(\omega), \qquad (2)$$

where $\delta f = f(-d_s/2) - f(d_s/2)$ and $\bar{f} = [f(-d_s/2) + f(d_s/2)]/2$ are the difference and average of the material property $f(z)$ across the interface, and $\sigma_{ij}$ and $\sigma_{ijk} = i(\epsilon_{jkl}T_{il} + \epsilon_{ikl}T_{jl}) + \omega S_{ijk}$ are the bulk conductivity tensors defined by $J_i = \sigma_{ij}E_j + \sum_{j,k}\sigma_{ijk}q_j E_k + O(q^2)$ for light wave vector $q$, where $T_{ij} = G_{ij} - \frac{1}{3}\delta_{ij}\sum_{k=1}^3 G_{kk} - \frac{i}{6}\omega\sum_{k,l=1}^3 \epsilon_{jkl}a'_{kli}$, and $S_{ijk} = (a'_{ijk} + a'_{jki} + a'_{kij})/3$[19,21].

## Table 1 | Symmetry properties of electromagnetic linear response functions

| Tensor | P | T | PT | Phenomena | Kerr | Faraday |
|---|---|---|---|---|---|---|
| $\chi_{ij}$ | + | + | + | Refraction and absorption | No | No |
| $\chi'_{ij}$ | + | − | − | Hall effect | Yes | Yes |
| $G_{ij}, a'_{ijk}$ | − | − | + | Optical magneto-electric effect | Yes | No |
| $G'_{ij}, a_{ijk}$ | − | + | − | Natural optical activity | No | Yes |

Response functions are defined in Eq. (1). The sign in the second column shows that parity of the response functions under spatial inversion $P$, time reversal $T$, and spacetime inversion $PT$. Kerr and Faraday in the last column indicate the optical rotation of the light polarization plane in reflection and transmission, respectively (i.e., the Kerr and Faraday effects). Kerr and Faraday effects are allowed when $T$ and $PT$ symmetry are broken, respectively[33,53,54].

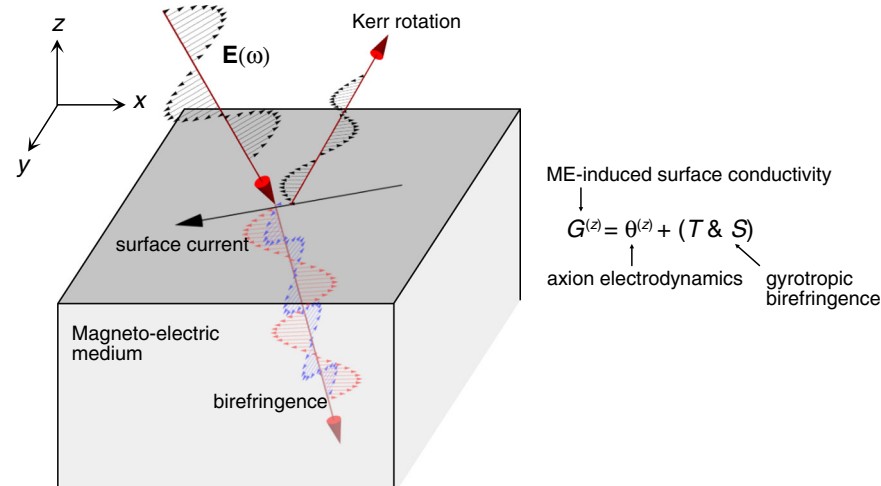

**Fig. 1 | Optical magneto-electric effect.** Magneto-electric (ME) coupling leads to different electrodynamics within the bulk and on the surface. Inside the magneto-electric medium, two linearly polarized light propagate with different speeds because the wave equation is modified by origin-independent magneto-electric coupling $T_{ij}$ and electric quadrupole susceptibility $S_{ijk}$ (this effect is called gyro-tropic birefringence[20]). On the surface, axion magneto-electric coupling $\theta^{(z)}$ comes into play additionally, contributing to the surface Hall conductivity. This is most readily seen by writing the action for optical axion electrodynamics allowing for a spatially varying $\theta$ parameter, $S_{OA} = (e^2/2\pi h)\int\theta(x)\mathbf{E}\cdot\mathbf{B}$. Integrating by parts in the presence of an interface along the $z$ direction where the axion angle jumps at the interface $\delta\theta$ and identifying the coefficient of $A_i$ with the surface current density we find $\mathbf{J} = -(e^2/2\pi h)\delta\theta\hat{z}\times\mathbf{E}$. Thus an electric field parallel to the surface sets up a current in the orthogonal direction along the surface, indicative of a surface Hall effect. $S_{OA}$ does not modify the propagation of electromagnetic fields within the bulk medium where $\theta$ does not vary spatially.

We neglect the $\delta E$ term because it is a bulk response whose contribution to the interface vanishes as $d_s \to 0$. The form of the surface current density in Eq. (2) suggests defining surface-sensitive magneto-electric coupling by

$$G_{ix}^{(z)}(\omega) = G_{ix}(\omega) - \frac{1}{2}\omega a'_{iyz}(\omega),$$
$$G_{iy}^{(z)}(\omega) = G_{iy}(\omega) + \frac{1}{2}\omega a'_{ixz}(\omega),$$
(3)

where the superscript $(z)$ explicitly shows that these quantities depend on the choice of the surface normal direction [$(z)$ and $(-z)$ are the same, though]. The $zz$ component of this surface-sensitive magneto-electric coupling is defined from the response of the surface charge. Using $\rho = -\nabla\cdot\mathbf{P} + \frac{1}{2}\partial_i\partial_j Q_{ij} + \ldots$, we obtain $\rho^s = G_{zi}B_i + \frac{\omega}{2}(a'_{yzx} - a'_{xzy})B_z + \frac{\omega}{2}a'_{zzy}B_x - \frac{\omega}{2}a'_{zzx}B_y + \ldots$ where the ellipsis includes the electric dipole term $\chi_{zi}E_i$, the symmetric $(\partial_j E_k + \partial_k E_j)$ terms, and higher-order multipole terms. From this, we define

$$G_{zz}^{(z)}(\omega) = G_{zz}(\omega) + \frac{1}{2}\omega\left[a'_{yxz}(\omega) - a'_{xyz}(\omega)\right]$$
(4)

as well as $G_{zx}^{(z)} = G_{zx} + \omega a'_{zzy}/2$ and $G_{zy}^{(z)} = G_{zy} - \omega a'_{zzx}/2$.

While various components of the magneto-electric coupling are related to the surface optical conductivity, there is a unique component that manifests itself only at the surface: the axion angle. We define the optical axion angle by the trace part of $G_{ij}^{(z)}$.

$$\theta^{(z)}(\omega) \equiv \pi\frac{2h}{e^2}\frac{1}{3}\sum_{i=1}^{3}G_{ii}^{(z)}(\omega).$$
(5)

To see that this only manifests itself at the surface rather than the bulk, we note that the bulk current response by the magneto-electric and electric quadrupole susceptibilities is determined only through $T_{ij}$ and $S_{ijk}$. Because of this, previous studies focusing on bulk magneto-electric effect did not capture optical axion electrodynamics[19,25]. See Supplementary Note 1 for more details on the bulk response. The surface-sensitive magneto-electric coupling is fully determined by these bulk-response quantities and the axion angle:

$$G_{xx}^{(z)}(\omega) = \frac{e^2}{2\pi h}\theta^{(z)}(\omega) + T_{xx}(\omega) - \frac{\omega}{2}S_{xyz}(\omega),$$
$$G_{yy}^{(z)}(\omega) = \frac{e^2}{2\pi h}\theta^{(z)}(\omega) + T_{yy}(\omega) + \frac{\omega}{2}S_{xyz}(\omega),$$
$$G_{zz}^{(z)}(\omega) = \frac{e^2}{2\pi h}\theta^{(z)}(\omega) + T_{zz}(\omega),$$
$$G_{ij}^{(z)}(\omega) = T_{ij}(\omega) - \frac{\omega}{2}S_{iik}(\omega)\epsilon_{jik}, \text{ for } i\neq j \text{ and } j\neq 3.$$
(6)

Since $T_{ij}$ is traceless, it does not contribute to the trace of $G_{ij}^{(z)}$. Note that $G_{ij}^{(z)}(\omega)$ transforms as a tensor under magnetic layer group actions but not under the full three-dimensional magnetic space group actions.

**Magneto-electric coupling with open boundaries in one direction**

Defining the combination $G_{ij}^{(z)}$ of $G_{ij}$ and $a'_{ijk}$ has an advantage in practical calculations as well as in the formulation. As $G_{ij}^{(z)}$ characterizes the surface current response which is measurable, it admits a gauge-invariant form in three-dimensional lattice systems with open boundaries along one direction, or in quasi-two-dimensional systems. This nice property is absent in the diagonal magneto-electric coupling $G_{ii}$, making it hard to calculate.

The bare magneto-electric coupling $G_{ij} = \partial_{Pi}/\partial_{Bj}$ and $a'_{ijk} = \partial Q_{jk}/\partial E_i$ at zero temperature have the following form according to linear response theory (see Methods).

$$G_{ij}(\omega) = \frac{V}{\hbar}\sum_{n,m}\frac{f_{nm}}{\omega_{mn}-\omega}\text{Re}\langle n|\hat{P}_i|m\rangle\langle m|\hat{M}_j|n\rangle,$$
$$a'_{ijk}(\omega) = -\frac{V}{\hbar}\sum_{n,m}\frac{f_{nm}}{\omega_{mn}-\omega}\frac{\omega_{mn}}{\omega}\text{Im}\langle n|\hat{P}_i|m\rangle\langle m|\hat{Q}_{jk}|n\rangle,$$
(7)

where $V$ is the volume of the system, $n, m$ are indices for energy eigenstates with eigenvalue $\hbar\omega_n$, $f_{nm} = f_n - f_m$ is the difference between the Fermi-Dirac distribution of the $n$ and $m$ states. $\hat{P}_i = -e\hat{r}^i/V$, $\hat{M}_j = -(e\epsilon_{jkl}\hat{r}^k\hat{v}^l/2 + \hat{m}_j^s)/V$, and $\hat{Q}_{jk} = -e\hat{r}^j\hat{r}^k/V$ are electric dipole,

magnetic dipole, and electric quadrupole density operators, respectively, where $\hat{r}$ and $\hat{v}$ are the position and velocity operators of electrons, and $\hat{m}^s$ is the spin magnetic moment operator. Equation (7) can be calculated for molecular systems[21,29], meaning finite systems with open boundary conditions along all directions, by using the real-space representation of $\hat{r}$ and the relation $\hat{v} = -i\hbar^{-1}[\hat{r}, \hat{H}]$, where $\hat{H}$ is the Hamiltonian of the system.

In periodic systems, however, $G_{ij}$ and $a'_{ijk}$ are not well defined separately because the position operator is not well-defined because the position is not uniquely defined. This manifests through the momentum-space representation of the position operator $\langle\psi_{mk'}|\hat{r}|\psi_{nk}\rangle = -\delta_{mn}i\partial_{\mathbf{k}}\delta_{\mathbf{k'k}} + \delta_{\mathbf{k'k}}\langle u_{mk}|i\partial_{\mathbf{k}}|u_{nk}\rangle$, whose diagonal matrix element is not well defined because of $\partial_{\mathbf{k}}\delta_{\mathbf{k'k}}$. On the other hand, the diagonal matrix elements of the position operator do not appear in the response functions that have a well defined physical meaning in periodic systems. $T_{ij}$ and $S_{ijk}$ are such examples that characterizes the bulk current response[19]. Since $G^{(z)}$ characterizes the surface response, one can expect that it is well defined in quasi-two-dimensional periodic systems.

After doing some algebra that we relegate to Methods, we can write $G^{(z)}_{ii}$'s in two-dimensional momentum space as

$$G^{(z)}_{xx}(\omega) = \frac{e^2}{\hbar V}\sum_{n\neq m,k_x,k_y}\frac{f_{nm}}{\omega_{mn}-\omega}\mathrm{Re}\left[r^x_{nm}\langle\psi_{m(k_x,k_y)}| - \frac{1}{2}(\hat{v}^y\hat{r}^z + \hat{r}^z\hat{v}^y) + \hat{m}^s_x|\psi_{n(k_x,k_y)}\rangle\right],$$

$$G^{(z)}_{yy}(\omega) = \frac{e^2}{\hbar V}\sum_{n\neq m,k_x,k_y}\frac{f_{nm}}{\omega_{mn}-\omega}\mathrm{Re}\left[r^y_{nm}\langle\psi_{m(k_x,k_y)}|\frac{1}{2}(\hat{v}^x\hat{r}^z + \hat{r}^z\hat{v}^x) + \hat{m}^s_y|\psi_{n(k_x,k_y)}\rangle\right],$$

$$G^{(z)}_{zz}(\omega) = \frac{e^2}{\hbar V}\sum_{n\neq m,k_x,k_y}\frac{f_{nm}}{\omega_{mn}-\omega}\mathrm{Re}\left[\frac{1}{2}\sum_{p:E_p\neq E_m}\left(r^z_{nm}r^x_{mp}v^y_{pn} - r^z_{np}r^x_{pm}v^y_{mn} - (x\leftrightarrow y)\right) + r^z_{nm}(m^s_z)_{mn}\right],$$

(8)

where $r^i_{mn} = \langle\psi_m|\hat{r}^i|\psi_n\rangle$ and $v^i_{mn} = \langle\psi_m|\hat{v}^i|\psi_n\rangle$ are matrix elements of the position and velocity operators, $|\psi_{n(k_x,k_y)}\rangle$ is the Bloch state, the subscripts $n,m$ and $p$ are band indices. While the position operator matrix element is not well defined in momentum space[30], the diagonal matrix elements of $\hat{r}^x$ and $\hat{r}^y$ do not appear in Eq. (8), so that the surface-sensitive magneto-electric coupling is well defined in two-dimensional momentum space. Combining equations in Eqs. (5) and (8), we can obtain the optical axion angle. In simple tight-binding models where $\hat{r}^z$ commutes with $\hat{v}^y$, $G^{(z)}_{xx}(0)$ reduces to the expression of $G_{xx}(0)$ derived by Liu and Wang[31]. To our knowledge, our formulas represent the first expressions for calculating the diagonal components of the optical magneto-electric coupling in crystalline (periodic) systems.

The form of $G^{(z)}_{xx}$ and $G^{(z)}_{yy}$ suggests that their orbital magneto-electric part may be interpreted as a real-space dipole of the Berry curvature. This idea works exactly for two-band systems in the limit of decoupled layer systems, where the matrix element part can be written as the product of the Berry curvature $F_{xy} = -2\mathrm{Im}[r^x_{12}r^y_{21}]$ and the $z$ component of the position eigenvalues $r^z_{11}$ or $r^z_{22}$. We apply this idea below to the case where the $z$ direction is also periodic.

### Intra-cell magneto-electric coupling and optical layer Hall conductivity

In fully periodic lattice systems where the $z$ direction is also periodic, it is hard to calculate the full orbital part of $G^{(z)}_{ii}(\omega)$ because $\hat{r}^z$ is not well defined in momentum space. Nevertheless, we can still define and calculate the magneto-electric coupling of the unit cell, which we call the intra-cell magneto-electric coupling. Note, this treatment will be necessarily approximate, in contrast to our previous discussion of the slab geometry, but provides an physical understanding for the results. For example, the use of the intra-cell magneto-electric coupling makes the relation concrete between the axion angle and the antiferromagnetism in inversion symmetric systems. Ultimately we must compare the results between this approach and the previous slab calculation as we do in another section below.

Let us begin by giving a physical intuition that a nonzero axion angle is natural in a fully compensated antiferromagnet[32]. To see this, recall the analogy between electric polarization in 1D and the axion theta angle $\theta$ in 3D. The former can be defined in a system free of net charge, by stacking alternate positive and negative charges. Similarly, if we stack alternate planes with Hall conductance $\pm g^H_{xy}$ such that the net Hall conductance vanishes, the axion magneto-electric coupling becomes well defined and is the analog of electric polarization of the alternating planes. In fact, with an applied magnetic field perpendicular to the planes, the induced polarization from having alternating charges in the $\pm g^H_{xy}$ layers, leads to a finite electric polarization which is readily calculated as $g^H_{xy}\delta a_z/a_z$, where $\delta a_z$ is spacing between alternating antiferromagnetic planes versus the vertical size of the unit cell $a_z$. The charge in each flux quantum area is $g_H(h/e)$. For an antiferromagnet where spacing between planes $= 1/2a_z$, this is just $\theta = 2\pi g^H_{xy}/2$.

To present a more detailed analysis, let us decompose the position operator into the intra-cell polarization and unit cell position parts by $r^z_{mn} = \mathbb{A}^z_{mn} + R^z_{mn}$. Namely,

$$\langle\psi_{mk'}|\hat{r}^z|\psi_{nk}\rangle = \delta_{\mathbf{k},\mathbf{k'}}\sum_{\beta,\alpha}\langle\psi_{mk}|\psi_{\beta k}\rangle\mathbb{A}^z_{\beta\alpha}(\mathbf{k})\langle\psi_{\alpha k}|\psi_{nk}\rangle$$
$$+ \sum_{\alpha,\mathbf{R}}\langle\psi_{mk'}|w_{\alpha\mathbf{R}}\rangle R^z\langle w_{\alpha\mathbf{R}}|\psi_{nk}\rangle$$

(9)

where $|w_{\alpha\mathbf{R}}\rangle$ is the Wannier state with the collective index $\alpha$ for spin and orbital (cf. $n$ and $m$ are band indices), $|\psi_{\alpha k}\rangle = N^{-1/2}\sum_{\mathbf{R}}e^{i\mathbf{k}\cdot\mathbf{R}}|w_{\alpha\mathbf{R}}\rangle$, and $\mathbb{A}^z_{\beta\alpha}(\mathbf{k}) = \sum_{\mathbf{R}}\langle w_{\beta 0}|\hat{r}^z|w_{\alpha\mathbf{R}}\rangle e^{i\mathbf{k}\cdot\mathbf{R}}$. The second term in Eq. (9) vanishes for $n\neq m$ as one can see by writing it as $-i\delta_{mn}\partial_{k^z}\delta_{\mathbf{k},\mathbf{k'}}$. This decomposition is independent of choosing the basis $\alpha$ within the unit cell, where each unit cell is labeled by a given lattice vector $\mathbf{R}$. However, the decomposition depends on the choice of the unit cell, which we discuss below.

The intra-cell polarization term, the first term in Eq. (9), defines the magneto-electric coupling of the unit cell. This intra-cell magneto-electric coupling depends on the choice of the unit cell. In our case, choosing a unit cell corresponds to fixing the value of the Wannier centers $\langle w_{\alpha 0}|\hat{r}^z|w_{\alpha 0}\rangle$, which is ambiguous by respective lattice translations of the Wannier states $|w_{\alpha 0}\rangle$. A physical interpretation of this multi-valuedness is similar to that of electric polarization[32]: The magneto-electric coupling depends on how we open the boundary, and there exists a preferred choice of the unit cell for each boundary condition (Fig. 2(a)).

The unit-cell position term has the form $\sum_{R_z}R^z\sigma^H_{xy}(R_z)$, where $\sigma^H_{xy}\equiv(\sigma_{xy}-\sigma_{yx})/2$ is the Hall conductivity, and $\sigma^H_{xy}(R^z) = -e^2\hbar^{-1}V^{-1}\sum_{n\neq m,k_x,k_y}f_{nm}\omega_{mn}/(\omega_{mn}-\omega)^{-1}\mathrm{Im}[r^x_{nm}r^y_{mn}P^{R^z}_{nn}]$, and $\hat{P}^{R^z} = \sum_{\alpha,R_x,R_y}|w_{\alpha\mathbf{R}}\rangle\langle w_{\alpha\mathbf{R}}|$ is the projection to $R^z$. This term is also multi-valued in periodic systems because the unit cell position $\mathbf{R}$ is not uniquely defined in periodic systems. This part, however, does not contribute to the magneto-electric coupling when the Hall conductivity of the unit layer vanishes, which is the case in $PT$-symmetric systems. When the boundary is introduced, the Hall conductivity of the surface unit layer can be nonzero even when the bulk Hall conductivity vanishes (Fig. 2(b)). This change of the $R^z$ term is the main source of the difference in the magneto-electric couplings between finite-size systems and periodic lattice systems. This effect is significant especially in the static limit of axion insulators, where the emergent Dirac surface states modify the low-frequency surface Hall conductivity in the order of $e^2/2h$.

As we derive below, in inversion-symmetric even-layer antiferromagnets, the intra-cell contribution to the magneto-electric coupling is simplified to the optical layer Hall conductivity. Namely, the optical

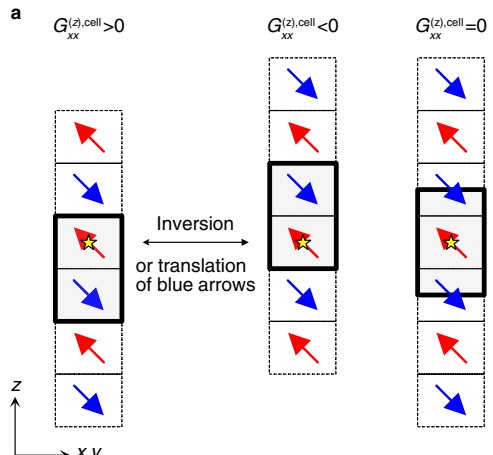

**Fig. 2 | Intra-cell and unit-cell-position contributions to optical magneto-electric coupling. a** Preferred choices of the unit cell with open boundaries. Arrows represent the average moment of a layer, where we bipartition the degrees of freedom within the unit cell into the upper and lower parts, each of which defines what we call a layer. The left-hand-side and middle even-layer systems are related by spatial inversion symmetry and also by the translation of the blue-arrow layers by a lattice constant if the deformation of the electronic states at the surface is neglected. The right-hand-side odd-layer system is inversion symmetric. The inversion centers are shown as yellow stars. **b** Hall conductivity of the unit cell in the presence of surface states. The vertical displacement $R_z$ of a unit cell times its net Hall conductivity contributes to diagonal magneto-electric coupling $G_{xx}^{(z)}$, $G_{yy}^{(z)}$, and $G_{zz}^{(z)}$.

axion angle is

$$\frac{e^2}{2\pi h}\theta^{(z)}(\omega) \approx \frac{1}{2}a_z\sigma_{xy}^{\mathrm{LH}}(\omega) + \sum_{R_z}R^z\sigma_{xy}^{H}(R_z) \quad \text{for inversion-symmetric AFM,}$$

(10)

where $a_z$ is the vertical lattice constant, and $\sigma_{xy}$ is a three-dimensional conductivity taking the unit of conductance (unit of $e^2/h$) divided by the length. As for the definition of layers, we note that there are two inversion-invariant values of the vertical displacement $z$ within $a_z$. We refer to the two quasi-two-dimensional bipartite regions centered at those two invariant $z$ values as layers (Fig. 2(a)). The layer Hall conductivity is then defined as $\sigma_{xy}^{\mathrm{LH}} = (\sigma_{xy}^{H,u} - \sigma_{xy}^{H,d})/2$ from the bulk optical Hall conductivity projected to the single layer $l = u, d$:

$$\sigma_{xy}^{H,l}(\omega) = -\frac{e^2}{\hbar V}\sum_{n\neq m, k_x, k_y}\frac{f_{nm}\omega_{mn}}{\omega_{mn} - \omega}\mathrm{Im}\left[r_{nm}^x r_{mn'}^y P_{n'n}^l\right],$$

(11)

where $P_{n'n}^l(\mathbf{k}) = \sum_{\alpha, \mathbf{R}; r_{\alpha,\mathbf{R}}^z \in l} {}_{\text{layers}}\langle\psi_{n\mathbf{k}}|w_{\alpha\mathbf{R}}\rangle\langle w_{\alpha\mathbf{R}}|\psi_{n'\mathbf{k}}\rangle$, $|w_{\alpha\mathbf{R}}\rangle$s are inversion-symmetric Wannier states, and $r_{\alpha,\mathbf{R}}^z = \langle w_{\alpha\mathbf{R}}|\hat{r}^z|w_{\alpha\mathbf{R}}\rangle$ is the Wannier center.

To obtain Eq. (10), note that the unit cell for even-layer systems is symmetric under the combination of inversion and a lattice translation of either the $u$ layer by $-a_z$ (or the $l$ layer by $+a_z$) (Fig. 2(a)). If we focus on the intra-cell part (the $\mathbb{A}$ part), the combined symmetry gives a constraint $\theta_{\mathbb{A}}^{(z)}(\omega) = -\theta_{\mathbb{A}}^{(z)}(\omega) - a_z\sigma_{xy}^{H,d}(\omega)$, where the last term is due to the transformation property of the axion angle $\delta\theta^{(z)}(\omega) = d_z\sigma_{xy}^{H}(\omega)$ under the translation by a vector $\mathbf{d}$. As we consider antiferromagnets with zero net Hall conductivity, such that $\sigma_{xy}^{H,d} = -\sigma_{xy}^{H,u} = -\sigma_{xy}^{\mathrm{LH}}$, we arrive at Eq. (10). Another useful way of understanding this result is to think of the electric polarization generated by applying a uniform magnetic field, and then the displacement of layers of one sign of the Hall effect obviously contributes to change in electric polarization, as we explained at the beginning of this section.

**Magneto-optic effects in fully compensated antiferromagnets**

The optical axion angle manifests directly through the magneto-optic Kerr effect. To gain some intuition for this, recall that the change in axion angle at the surface leads to a surface Hall effect $\sigma_{xy}^s = (\delta\theta/2\pi)e^2/h$. Clearly, such a surface Hall conductance will lead to a Kerr effect. We present a more systematic analysis below.

Let us consider the reflection at the single interface between two media 1 and 2 (Fig. 3). For simplicity, we assume that both media have $M_xT$, $C_{3z}$, and $PT$ symmetries, which are shared by magneto-electric materials $Cr_2O_3$ and $MnBi_2Te_4$. We also assume normal incidence (i.e., light is incident along $-\hat{z}$). Then there is no bire-fringence because of the symmetry we require, so the propagation of light within each medium is then characterized by a single complex-valued refractive index $n_\mu$, where $\mu = 1, 2$ labels the two media (See Eq. (6)).

The electric field in medium 1 consists of incident and reflected fields $\mathbf{E}_i$ and $\mathbf{E}_r$ while that in medium 2 is the transmitted field $\mathbf{E}_t$:

$$\mathbf{E}_1 = \mathbf{E}^i + \mathbf{E}^r \equiv (1 + r)\mathbf{E}^i,$$
$$\mathbf{E}_2 = \mathbf{E}^t \equiv t\mathbf{E}^i,$$

(12)

where $r$ and $t = 1 + r$ are $2 \times 2$ Jones matrices for reflection and transmission, respectively. $\mathbf{E}_1 = \mathbf{E}_2$ at the interface because electric fields are continuous by Faraday's law, because we consider normal incidence such that electric fields are parallel to the interface. The contribution from the surface conductivity is encoded in the boundary condition of the magnetic field at the interface.

$$\mathbf{B}^t = \mathbf{B}^i + \mathbf{B}^r + \mu_0\hat{z}\times\mathbf{j}^s$$

(13)

Here, we consider only the surface currents induced from the bulk, i.e., $\hat{z}\times\mathbf{j}^s = \sigma_{xy}^s\mathbf{E}$, where $\sigma_{xy}^s = G_{xx}^{(z),\mu=2} - G_{xx}^{(z),\mu=1}$. By solving the boundary condition equations, we obtain

$$r_{xx} = \frac{(n_1 - n_2)(n_1 + n_2) - (\mu_0 c\sigma_{xy}^s)^2}{(\mu_0 c\sigma_{xy}^s)^2 + (n_1 + n_2)^2},$$
$$r_{xy} = -\frac{2n_1(\mu_0 c\sigma_{xy}^s)}{(\mu_0 c\sigma_{xy}^s)^2 + (n_1 + n_2)^2}.$$

(14)

Magneto-electric coupling appears in the reflective Jones matrix through the surface conductivity, which is a manifestation that the Kerr effects here are surface phenomena.

The complex Kerr angle is defined by $\phi_K = \varphi_K + i\eta_K = \tan^{-1}(r_{xy}/r_{xx})$. Its real part measures the rotation of the light

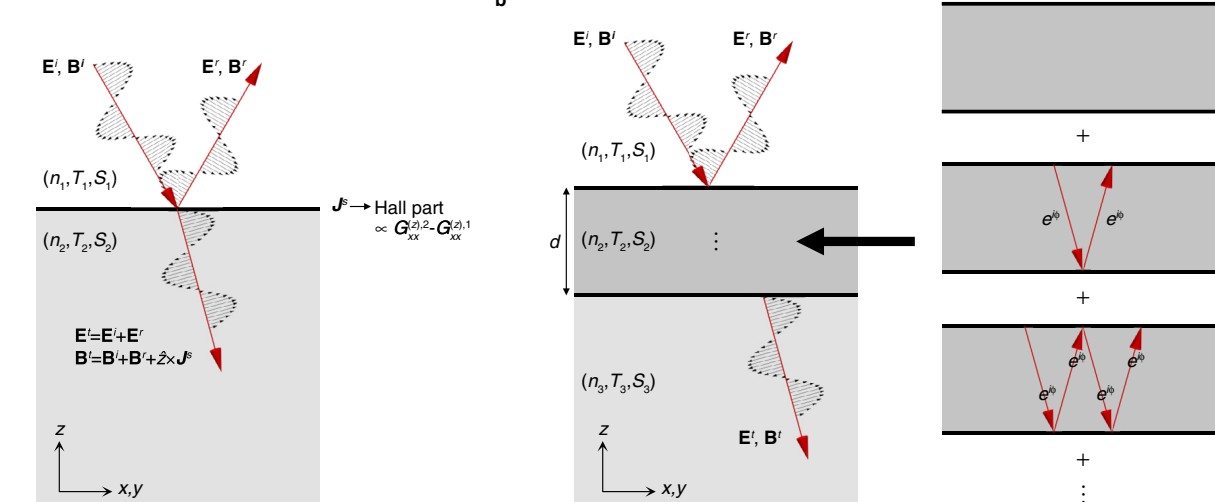

**Fig. 3 | Reflections by *PT*-symmetric magneto-electric media.** We assume *PT* symmetry to exclude natural optical activity, for simplicity. **a** Single interface. The light propagation within each medium is determined by the refractive index $n$ and origin-independent magneto-electric coupling $T_{ij}$ and electric quadrupole susceptibility $S_{ijk}$ tensors. We consider the case where no birefringence occurs such that the light propagation is described by a unique refractive index in each medium. **b** Double interfaces. Infinite reflections occur between the top and bottom surfaces of medium 2. Light obtains the complex phase $\phi = kd = n_2\omega d/c$ while propagating the distance $d$ in medium 2.

polarization plane while its imaginary part measures the circular dichroism, i.e., the intensity imbalance between the reflected left and right circularly polarized light. The Kerr angle is $O(\mu_0 c\sigma^s_{xy})$ in general, but it can be much enhanced when $n_1 \approx n_2$ because of the suppression of $r_{xx}$.

When the sample thickness is much larger than the wavelength of light, it is enough to suppose that reflection occurs at a single interface. However, when thickness $d$ is comparable to or less than wavelength $\lambda$, which is the case particularly relevant for thin films, we need to consider the response from the whole sample including top and bottom surfaces (cf. When the photon energy is smaller than the band gap, double interfaces can be relevant even for $d \gg \lambda$ for an insulating medium[14] because then light incident on the top reaches the bottom without attenuation.).

To study this case, we consider three media with refractive index $n_\mu$, where $\mu = 1, 2, 3$, as shown in Fig. 3b. We again assume $M_xT$, $C_{3z}$, and *PT* symmetries for each media. The Jones reflection matrix of the sample for light incident from medium 1 is then the infinite sum of multiple reflections.

$$r = r_T + e^{2i\phi}(1+r'_T)r_B(1 - e^{2i\phi}r'_Tr_B)^{-1}(1+r_T),\tag{15}$$

where we use $t_{T,B} = 1 + r_{T,B}$ and $t'_T = 1 + r'_T$, and $\phi = n_2\omega d/c$ is the complex-valued phase obtained by the one-way propagation through the sample thickness $d$. Here, the subscripts $T$ and $B$ indicate the top and bottom of the sample, and the prime indicates the process where the light is incident to the top surface from below (we follow the notation in ref. 13). See Methods for the expression of Jones matrices $r_T$, $r'_T$, and $r_B$. Similarly, for transmission,

$$t = (1+r_B)(1 - e^{2i\phi}r'_Tr_B)^{-1}e^{i\phi}(1+r_T).\tag{16}$$

Note that $t \neq 1 + r$ when $\phi \neq 0$.

Considering the case where only medium 2 is magneto-electric, we set $G^{(z),1}_{xx} = G^{(z),3}_{xx} = 0$ and $G^{(z),2}_{xx} \equiv G^{(z)}_{xx} \neq 0$. We first consider $|\delta n| \ll |\phi|$, where $\delta n \equiv n_3 - n_1$. The Kerr and Faraday rotation angles are then given by

$$\tan\phi_K = \frac{2n_1\mu_0 cG^{(z)}_{xx}}{n_2^2 - n_1^2 + (\mu_0 cG^{(z)}_{xx})^2} + O(\delta n),$$

$$\tan\phi_F = -\frac{\mu_0 cG^{(z)}_{xx}\sin\phi}{\left(n_1^2 + n_2^2 + (\mu_0 cG^{(z)}_{xx})^2\right)\sin\phi + 2in_1n_2\cos\phi}\delta n + O(\delta n^2),$$

$$\tag{17}$$

where $\tan\phi_F = t_{xy}/t_{xx}$. A nonzero Faraday rotation requires $\delta n \neq 0$ because *PT* symmetry needs to be broken[33]. On the other hand, the Kerr rotation can be nonzero with $\delta n = 0$ because it is compatible with *PT* symmetry (Table 1). The Kerr rotation is independent of $\phi$ in the leading order of $\delta n$ when $\delta n$ is sufficiently small as if the response comes from the top surface only, while it actually comes from multiple reflections between the top and bottom surfaces. In this limit, the way the Kerr effect goes away is highly nontrivial as the thickness goes to zero (i.e., $|\phi| \to 0$ with $|\phi| \gg |\delta n|$). The Kerr angle stays constant, but the amplitudes of the reflected signals ultimately vanish. Therefore, a finite Kerr effect can be observed from a thin film when optical equipment is highly sensitive.

However, a nontrivial $\phi$ dependence appears when $\phi$ is the smallest parameter ($|\delta n| \gg |\phi|$), where we have

$$\tan\phi_K = \frac{4in_1n_3\mu_0 cG^{(z)}_{xx}}{n_2(n_1^2 - n_3^2)}\phi + O(\phi^2),$$

$$\tan\phi_F = -\frac{i(n_1 - n_3)\mu_0 cG^{(z)}_{xx}}{n_2(n_1 + n_3)}\phi + O(\phi^2),$$

$$\tag{18}$$

which show that both Kerr and Faraday rotation vanish for $\phi = 0$. Therefore, a nonzero $\phi$ is necessary for the Kerr effect as well as the Faraday effect in fully compensated antiferromagnets. Note that, while the Kerr angle in Eq. (17) is independent of $\phi$, it applies only when $|\phi| \gg |\delta n|$.

The crossover between two regimes respectively described by Eqs. (17) and (18) occurs when $|\phi| \sim |\delta n|$. The corresponding wavelength

scale

$$\lambda \sim \lambda^* \equiv 2\pi \left| \frac{n_2}{\delta n} \right| d \qquad (19)$$

is much larger than the sample thickness $d$ when $|\delta n| \ll 1$. This is remarkable because one might naively expect that, because the response from the spatially separated top and bottom layers are not resolved when $\lambda \gg d$, the Kerr effect is vanishingly small in that regime and scales linearly with $d/\lambda$. However, our analysis shows that a much stricter $\lambda \gg \lambda^*$ is required for the suppression of the Kerr angle. To explain how this works, we note again that the Kerr angle and the amplitude of the Kerr rotated signal can behave differently. While the amplitude of the Kerr rotated signal ($\propto r_{xy}$) is indeed suppressed for $\lambda \gg d$, the amplitude of the non-rotated signal ($\propto r_{xx}$) is also suppressed in the same limit. Their suppression at large wavelengths compensate each other to keep the ratio $\phi_K = r_{xy}/r_{xx}$ as long as $\lambda$ is smaller than a larger length scale $\lambda^*$, above which the Kerr angle ultimately gets suppressed.

In thin film axion insulators, Eq. (18) is typically more relevant at photon energies below the surface gap. The surface gap of experimentally realized axion insulators are about 50 meV[16,34,35]. For example, if we take $n_2 = 5$ and $d = 1$ nm, we obtain a very small value $|\phi| \leq 1.27 \times 10^{-3}$ at $\hbar\omega \leq 50$ meV, which is typically smaller than $|\delta n|$. On the other hand, in the infrared and visible regime where the photon energy is in the order of eV, equation (17) can become relevant. We demonstrate this in the following section.

## Model calculations

Let us apply our theory to study the optical axion electrodynamics in MnBi$_2$Te$_4$. MnBi$_2$Te$_4$ is the only stoichiometric compound that experimentally realizes the intrinsic antiferromagnetic axion insulator[34,35], which has now become an attractive platform for studying axion magneto-electric effects[10–12,36–39]. As it is a layered antiferromagnet, its few-layer behavior and layer number dependence is also of interest[31,40].

Here we calculate its magneto-optic properties based on the low-energy model in refs. 36,41. The goal of our calculations here is to cement the validity of our new theory by providing a concrete model example as well as to understand qualitative features (e.g., the dominance of the axion contribution and the significant Kerr and negligible Faraday effects) of the magneto-optic response in MnBi$_2$Te$_4$. Our model is expected to quantitatively capture the low-energy properties of the material. On the other hand, at photon energies much larger than the band gap, a precise quantitative calculation of the magneto-optical spectrum will require a model, like the full ab-initio model, that captures all the significant optical transitions involving those states neglected in our model. With this in mind, we consider a nearest-neighbor tight-binding Hamiltonian on a layer-stacked triangle lattice

$$\hat{H} = \sum_{i,\alpha,\beta} \hat{c}_{i\alpha}^\dagger (h_0)_{\alpha\beta} \hat{c}_{i\beta} - \sum_{\langle i,j \rangle,\alpha,\beta} \hat{c}_{i\alpha}^\dagger t_{\alpha\beta}^{ij} \hat{c}_{j\beta}, \qquad (20)$$

where $i,j$ are the site indices, $\langle i,j \rangle$ means that the summation is over nearest neighbors, and $\alpha, \beta = 1, \ldots, 4$ run over two spin and two orbital degrees of freedom at each site.

As the non-magnetic state has space group $R\bar{3}m$ (No. 166), we impose time reversal $T = is_y K$, inversion $P = \tau_z$, and threefold $C_{3z} = \exp(-i\pi t s_z/3)$ and twofold $C_{2x} = -is_x$ rotational symmetries, where $s_i$ and $\tau_i$ are Pauli matrices for spin and orbital, respectively. The onsite Hamiltonian satisfying all the symmetries of the nonmagnetic state is $h_0 = e_0 + e_5\tau_z$. Along the $z$ direction, the nearest-neighbor hopping matrices are $T_4 \equiv t^{j+\mathbf{a}_4,j} = t_0^z + it_3^z s_z \tau_x + t_5^z \tau_z$, where $\mathbf{a}_4 = (0, 0, a_z)$, $a_z$ is the out-of-plane lattice parameter. For the in-plane directions, the hopping matrices are $t^{j+\mathbf{a}_1,j} = t_0 + it_1 s_x \tau_x + it_4 \tau_y + t_5 \tau_z = (t^{j,j+\mathbf{a}_1})^\dagger$, $t^{j+\mathbf{a}_2,j} = C_{3z} T_1 C_{3z}^{-1}$, and $t^{j+\mathbf{a}_3,j} = C_{3z} T_2 C_{3z}^{-1}$, where $\mathbf{a}_1 = (a, 0, 0)$, $\mathbf{a}_2 = C_{3z}\mathbf{a}_1$,

$\mathbf{a}_3 = C_{3z}\mathbf{a}_2$, $a$ is the in-plane lattice parameter (see Methods for further details).

We consider the effect of the layer-alternating (i.e., A-type) antiferromagnetism by adding $(-1)^{n-1} m\sigma_z$ to $h_0$, where $n$ is the layer index. While this term breaks time reversal symmetry, inversion symmetry remains the symmetry of the lattice system. However, finite even-layer systems break inversion symmetry and have non-zero axion angle according to Eq. (10).

Figure 4a shows the orbital part of the axion angle calculated with the tight-binding parameters of MnBi$_2$Te$_4$ derived in ref. 36 (We make a momentum-dependent overall energy shift to obtain an insulating filling at half filling as we describe in Methods. The band structure and electric susceptibility are shown in Supplementary Figs. 1 and 2.). At high energies above 1 eV, the axion angle is well approximated by the optical layer Hall conductivity for any number of layers. However, the axion angle deviates significantly from the optical layer Hall conductivity as the photon energy gets lower below 1 eV. The deviation at the low energy increases with the number of layers because the surface massive Dirac fermion is then more localized at the surfaces and increases the second term in Eq. (10), making the static axion angle reach the quantized value $\theta = \pi$.

$T_{xx}$ and spin magneto-electric coupling are much smaller than the axion angle, as shown in Fig. 4b,c, respectively. This is consistent with our expectation that these origin-independent magneto-electric couplings are strongly suppressed in systems with local inversion symmetry. Furthermore, they decrease inversely with the number of layers $N_l$[31], because only $O(1/N_l)$ portion of layers near the top and bottom generates a nontrivial response. This contrasts to the case with a finite $T_{xx}$ for $N_l \to \infty$, where the response is coming from $O(N_l)$ layers.

As $T_{xx}$ is relatively small, optical axion electrodynamics dominates the Kerr and Faraday effects in this system. Figure 5 shows the Kerr and Faraday rotation angles calculated with the magneto-electric coupling. We consider the case where the model system (medium 2) is encapsulated by medium 1 and medium 3, having frequency-independent refractive indices $n_1 = 2.2$ and $n_3 = 2.4$, respectively, corresponding to those of the hexagonal Boron nitride and diamond at photon energy around 1 eV[42,43]. The calculated Kerr rotation angle $\varphi_K$ is about 0.02° at photon energies larger than 1 eV, which is about one order of magnitude smaller than $\varphi_K \lesssim 1$° in typical ferromagnets although our antiferromagnetic system has zero net magnetic moment. The Kerr angle in real MnBi$_2$Te$_4$ can even be much enhanced because of the contributions from higher-energy bands that we do not include here. As we consider $n_1 \neq n_3$, the Faraday rotation is nonzero because the cancellation between the top and bottom surfaces is incomplete. The Faraday effect is two orders of magnitude weaker than the Kerr effect.

## Discussion

Our theory of optical axion electrodynamics fills a crucial missing piece in the macroscopic theory of magneto-optic effects in antiferromagnets developed mostly by Graham and Raab[21,25–27]. They used only origin-independent $T_{ij}$ and $S_{ijk}$ to ensure physically meaningful results such as the consistency with the reciprocal relations. However, our approach shows that we can include the origin-dependent axion magneto-electric coupling in the theory, and it is precisely the axion angle that controls the Kerr effect in antiferromagnets with local inversion symmetry. As we show by using the low-energy tight-binding model of MnBi$_2$Te$_4$, the omission of the axion electrodynamics can underestimate the Kerr effect by orders of magnitudes. In general, the same suppression of $T_{ij}$ and $S_{ijk}$ is expected in systems with bulk symmetries that reverses an odd number of spacetime coordinates. As long as those symmetries are broken at the surfaces, the axion magneto-electric coupling is not much affected by the symmetries, such that axion electrodynamics dominates the response.

In fact, the trace part of the magneto-electric coupling was included in the study by Hornreich and Shtrikman[20] prior to

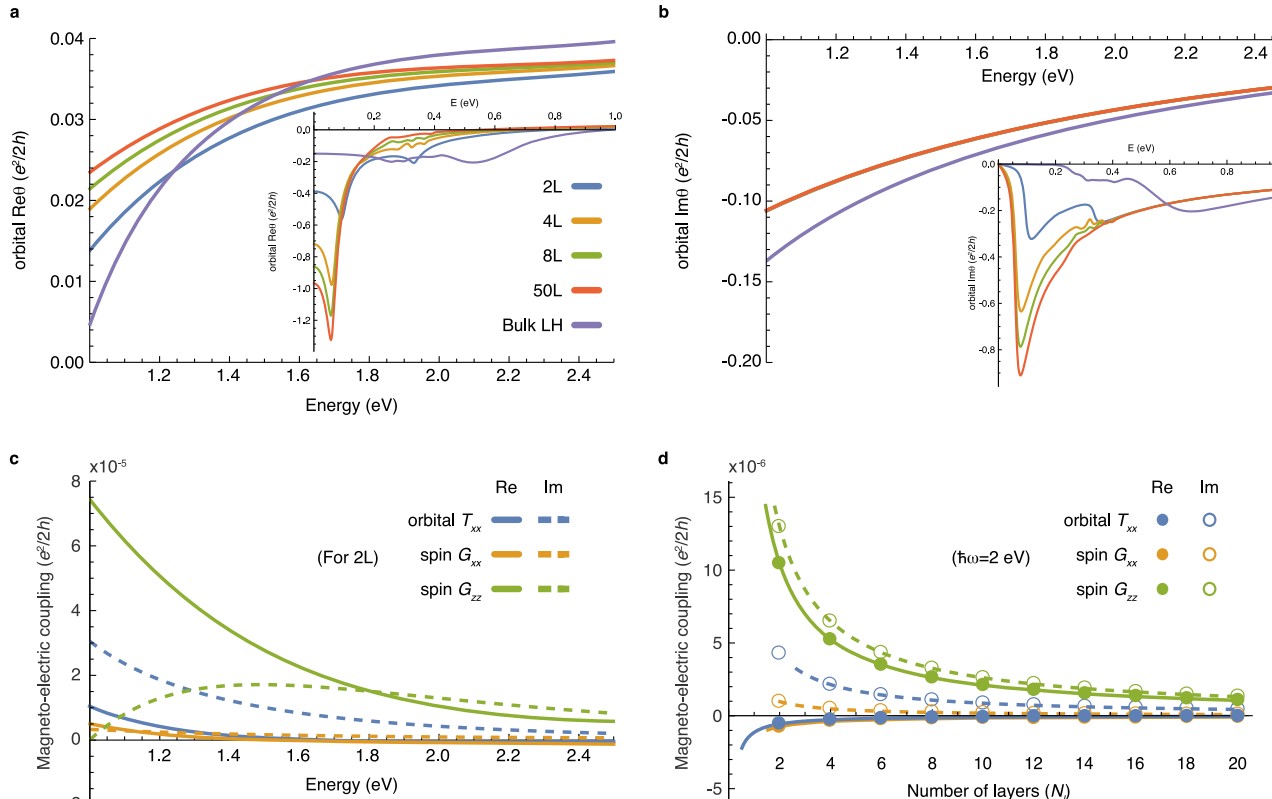

**Fig. 4 | Optical magneto-electric coupling in the low-energy tight-binding model of MnBi₂Te₄. a** and **b** Real and imaginary part of the orbital magneto-electric axion angle. Bulk LH indicates the optical layer Hall conductivity calculated with periodic boundary conditions. Bulk LH approaches the exact axion magneto-electric coupling as photon energy increases. Orbital magneto-electric $T_{xx}$ and the spin magneto-electric coupling. Spin parts contribute to axion angle and $T_{xx}$ through $\theta^{(z),s} = (2G_{xx}^s + G_{zz}^s)/3$ and $T_{xx}^s = (G_{xx}^s - G_{zz}^s)/3$. **c** Spectra for two layers. These are three orders of magnitudes smaller than the orbital axion magneto-electric coupling. **d** Layer-number dependence at photon energy $\hbar\omega = 2$ eV. Solid and dashed lines are curves fitted with $a/N_l + b$ form, where $a$ and $b$ are fitting parameters. All optical response functions are calculated with $\gamma = 10$ meV to broaden the resonance through $\omega \to \omega + i\gamma$.

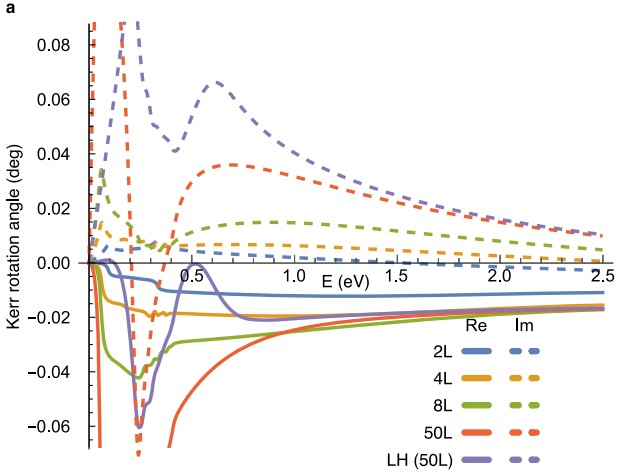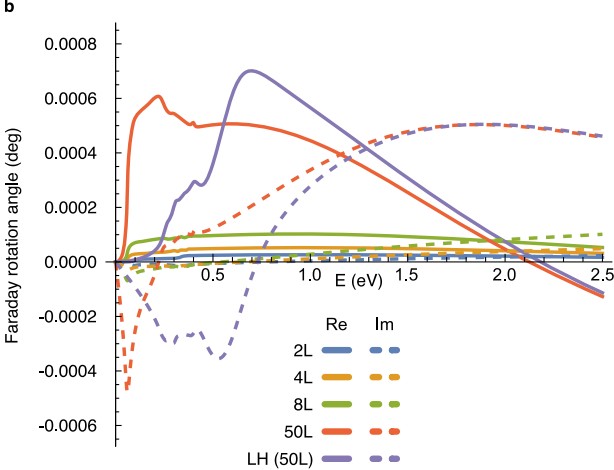

**Fig. 5 | Kerr and Faraday rotation angles in the low-energy tight-binding model of MnBi₂Te₄. a** Complex Kerr rotation angle $\phi_K = \tan^{-1}(r_{xy}/r_{xx})$. **b** Complex Faraday rotation angle $\phi_F = \tan^{-1}(t_{xy}/t_{xx})$. The real and imaginary parts of the complex angles are shown as solid and dashed lines, respectively. We consider the effect of both top and bottom surfaces by using Eqs. (15) and (16) with the refractive indices $n_1 = 2.2$ and $n_3 = 2.4$ for the capsulating media. The complex refractive index of the model itself is obtained from the electric dipole susceptibility through $n_2(\omega) = \sqrt{1 + \chi_{xx}(\omega)/\epsilon_0}$. For LH, we calculate the axion angle from the layer Hall conductivity with periodic boundary conditions and use the thickness of 50 layers for $\phi = n_2\omega d/c$. All optical response functions are calculated with $\gamma = 10$ meV to broaden the resonance through $\omega \to \omega + i\gamma$.

refs. 21,25–27, However, their estimate of the effect was four orders of magnitudes smaller than the value experimentally observed subsequently[44]. This inconsistency lead to the appearance of theories based on different approaches[24–27], all of which do not include the axion electrodynamics. Our introduction of the surface-sensitive magneto-electric coupling and the study of double interfaces allow for precise quantitative understanding of the Kerr effect including the axion electrodynamics, especially for thin films.

While we focus on the Kerr effect due to macroscopic magneto-electric coupling for fully compensated antiferromagnets, there is another microscopic mechanism based on the spatial modulation (phase change as well as the attenuation) of the electric field $E(z) = E_0 e^{ik_z z}$[24], where $k_z = n\omega/c$ is complex valued. However, the microscopic mechanism produces only minor effects. To see this, let $\phi_{K,0}$ be the Kerr rotation angle by a single layer with net magnetization in A-type antiferromagnet. Considering the reflection of each layer as well as the complex phase rotation during the propagation, one obtains the Kerr angle $\phi_K = \phi_{K,0}(1 - e^{2ik_z a_z} + e^{4ik_z a_z} - e^{6ik_z a_z} + \ldots)/(1 + e^{2ik_z a_z} + e^{4ik_z a_z} + e^{6ik_z a_z} + \ldots) = -ik_z a_z \phi_{K,0} + O((k_z a_z)^2)$ for an even number layers[24], where $a_z$ is the layer spacing. This Kerr angle is smaller than the axion-induced $\phi_K \approx \phi_{K,0}$ because $a_z \ll \lambda$ for the wavelength down to the UV regime.

In the static limit, both Faraday and Kerr effects are often considered as manifestations of the axion electrodynamics[8,13,14]. It is because the systems under consideration have finite net Hall conductivity. The same sign of the Hall conductivity is induced on the top and bottom surfaces of a $Z_2$ topological insulator by either external magnetic fields[8,13] or coupling to ferromagnets[14]. The main focus of those studies is the manifestation of the half-quantized surface Hall conductivity, rather than the magneto-electric response of antiferromagnets.

An open question we leave for future studies is to formulate a quantum geometric theory of optical axion electrodynamics in periodic systems, generalizing the Chern-Simons integral in the static limit. A drawback of calculating intra-cell optical magneto-electric coupling through Eq. (9) (or calculating the layer Hall conductivity) is that it does not capture the topological magneto-electric effect because we drop the second term in Eq. (10). A unified formula that captures both intra-cell optical magneto-electric coupling and topological magneto-electric effect is desired, and it is likely to require extending the quantum geometric formulation for electric dipole moments[45] to magnetic dipole and electric quadrupole moments and defining a proper optical Chern-Simons integral. However, this may not be feasible, in which case we are forced to work with quasi-two-dimensional systems.

Finally, we note that the optical axion angle we define should be distinguished from the dymanical axion fields[46]. In our optical axion electrodynamics based on linear response theory, the effective action $S_{OA} \propto \int d\omega d\mathbf{x} \theta(\omega, \mathbf{x}) \mathbf{E}(\omega, \mathbf{x}) \cdot \mathbf{B}(\omega, \mathbf{x})$ for non-absorptive media describes the propagation of the electromagnetic fields modified by elastic scattering by the medium. The optical axion angle is a ground-state property, which is non-dynamical. On the other hand, the dynamical axion field interacts spacetime-locally with the electromagnetic fields through $S_{dynamic} \propto \int dt d\mathbf{x} \theta(t, \mathbf{x}) \mathbf{E}(t, \mathbf{x}) \cdot \mathbf{B}(t, \mathbf{x})$. This describes Raman scattering where the energy or momentum of the incoming electromagnetic field is tranfered to the dynamical axion field. The interplay between the two distinct phenomena is an interesting research direction.

## Methods
### Generalization to include natural optical activity
**Electromagnetic multipole moments.** Let us consider electric dipole $P_i$, electric quadrupole $Q_{ij}$, and magnetic dipole $M_i$ moment densities induced by electric and magnetic fields[21]:

$$
\begin{aligned}
P_i &= P_i^0 + \chi_{ij} E_j + \frac{1}{2} a_{ijk} \nabla_k E_j + G_{ij} B_j \\
&\quad + \omega^{-1} \left[ \chi'_{ij} \dot{E}_j + \frac{1}{2} a'_{ijk} \nabla_k \dot{E}_j + G'_{ij} \dot{B}_j \right] \ldots \\
Q_{ij} &= Q_{ij}^0 + \mathfrak{a}_{ijk} E_k + \omega^{-1} \mathfrak{a}'_{ijk} \dot{E}_k \ldots \\
M_i &= M_i^0 + \mathfrak{G}_{ij} E_j + \omega^{-1} \mathfrak{G}'_{ij} \dot{E}_j \ldots
\end{aligned}
\tag{21}
$$

for monochromatic electromagnetic fields in time domain, where $P_i^0$, $Q_{ij}^0$, and $M_i^0$ are permanent multipole moments, $\mathfrak{a}_{ijk} = a_{kij}$, $\mathfrak{a}'_{ijk} = -a'_{kij}$,

$\mathfrak{G}_{ij} = G_{ji}$, $\mathfrak{G}'_{ij} = -G'_{ji}$. The ellipsis "..." indicates electric-octupole/magnetic-quadrupole or higher-order multipole contributions that we neglect here. Here, the primed susceptibility tensors transform oppositely under time reversal compared to the non-primed ones. For example, while $\chi_{ij}$ is even under time reversal, $\chi'_{ij}$ is odd under time reversal.

Electromagnetic multipole moment densities are defined by[21]

$$
\begin{aligned}
\hat{P}_i &= -e\hat{r}^i/V, \\
\hat{Q}_{ij} &= -e\hat{r}^i \hat{r}^j/V, \\
\hat{M}_i &= \frac{1}{4} \epsilon_{ijk} \left( \hat{r}^j \hat{J}_k^{orb} - \hat{J}_j^{orb} \hat{r}^k \right) + \hat{M}_i^{spin} \\
&= \hat{M}_i^{orb} + \hat{M}_i^{spin},
\end{aligned}
\tag{22}
$$

where we split the orbital magnetic and spin parts, which respectively originates from the minimal coupling $\nabla \to \nabla + ie\mathbf{A}$ and the explicit dependence on $\mathbf{B}$ independent of the minimal coupling.

$$
\begin{aligned}
\hat{J}_i^{orb} &= \frac{1}{V} \frac{\partial \hat{H}}{\partial A_i} |_{\mathbf{B} \text{ fixed}}, \\
\hat{M}_i^{spin} &= -\frac{1}{V} \frac{\partial \hat{H}}{\partial B_i} |_{\mathbf{A} \text{ fixed}}.
\end{aligned}
\tag{23}
$$

**Surface-sensitive magneto-electric coupling.** By generalizing the procedure in the main text to include both natural optical activity and gyrotropic birefringence, we define

$$
\begin{aligned}
\tilde{G}_{ix}^{(z)}(\omega) &= \tilde{G}_{ix}(\omega) - \frac{i}{2} \omega \tilde{a}_{iyz}(\omega), \\
\tilde{G}_{iy}^{(z)}(\omega) &= \tilde{G}_{iy}(\omega) + \frac{i}{2} \omega \tilde{a}_{ixz}(\omega), \\
\tilde{G}_{zz}^{(z)}(\omega) &= \tilde{G}_{zz}(\omega) - \frac{i}{2} \omega \left[ \tilde{a}_{zxy}(\omega) - \tilde{a}_{zyx}(\omega) \right]
\end{aligned}
\tag{24}
$$

from the surface response

$$
\begin{aligned}
j_x^s &= \sum_{i=1}^{3} \left( \mathfrak{G}_{yi} - \frac{i}{2} \omega \tilde{a}_{xzi} \right) E_i + \ldots, \\
j_y^s &= -\sum_{i=1}^{3} \left( \mathfrak{G}_{xi} + \frac{i}{2} \omega \tilde{a}_{yzi} \right) E_i + \ldots, \\
\rho^s &= \tilde{G}_{zi} B_i - \frac{i}{2} \omega (\tilde{a}_{zxy} - \tilde{a}_{zyx}) B_z - \frac{i}{2} \omega \tilde{a}_{yzz} B_x + \frac{i}{2} \omega \tilde{a}_{xzz} B_y + \ldots,
\end{aligned}
\tag{25}
$$

where $\tilde{G}_{ij} = G_{ij} - iG'_{ij}$, $\tilde{\mathfrak{G}}_{ij} = G_{ji} + iG'_{ji}$, $\tilde{a}_{ijk} = a_{ijk} - ia'_{ijk}$, and $\tilde{\mathfrak{a}}_{ijk} = a_{kij} + ia'_{kij}$. We can also define

$$
\tilde{G}_{zi}^{(z,2)}(\omega) = \tilde{G}_{zi}(\omega) - \frac{i}{2} \omega \sum_k \tilde{a}_{kzz}(\omega) \epsilon_{kzi} \text{ for } i = x, y,
\tag{26}
$$

which is distinguished from $\tilde{G}_{zi}^{(z)}(\omega)$. However, we focus on $\tilde{G}_{zi}^{(z)}(\omega)$ because we are interested in current generation rather than the charge fluctuation on the surface. We decompose $\tilde{G}_{ij}^{(z)}$ into

$$
\begin{aligned}
\tilde{G}_{xx}^{(z)}(\omega) &= \frac{e^2}{2\pi h} \theta^{(z)}(\omega) + \tilde{T}_{xx}(\omega) - \frac{\omega}{2} \left[ S_{xyz}(\omega) + i\Gamma_{xyz}(\omega) \right], \\
\tilde{G}_{yy}^{(z)}(\omega) &= \frac{e^2}{2\pi h} \theta^{(z)}(\omega) + \tilde{T}_{yy}(\omega) + \frac{\omega}{2} \left[ S_{xyz}(\omega) + i\Gamma_{xyz}(\omega) \right], \\
\tilde{G}_{zz}^{(z)}(\omega) &= \frac{e^2}{2\pi h} \theta^{(z)}(\omega) + \tilde{T}_{zz}(\omega), \\
\tilde{G}_{i\neq j}^{(z)}(\omega) &= \tilde{T}_{ij}(\omega) + \frac{\omega}{2} \epsilon_{zij} \left[ S_{iiz}(\omega) + i\Gamma_{iiz}(\omega) \right], \text{ for } i, j \in 1, 2, \\
\tilde{G}_{zi}^{(z)}(\omega) &= \tilde{T}_{zj}(\omega) + \frac{\omega}{2} \sum_{k=1}^{3} \epsilon_{kzi} \left[ S_{zzk}(\omega) - ia_{kzz} \right], \text{ for } i \in 1, 2,
\end{aligned}
\tag{27}
$$

where

$$\Gamma_{ijk} = a_{ijk} + a_{jik} - a_{kij},$$

$$\tilde{T}_{ij} = G_{ij} - \frac{1}{3}\delta_{ij}\sum_{k=1}^{3} G_{kk} - \frac{i}{6}\omega\sum_{k,l=1}^{3}\epsilon_{jkl}a'_{kli} - i\left(G'_{ij} - \sum_{k,l=1}^{3}\epsilon_{jkl}\frac{1}{2}\omega a_{kli}\right),$$

(28)

and $\theta^{(z)}(\omega) = 2\pi h e^{-2}\sum_{i=1}^{3} G_{ii}^{(z)}(\omega)/3$ and $S_{ijk}(\omega) = \frac{1}{3}[a'_{ijk}(\omega) + a'_{jki}(\omega) + a'_{kij}(\omega)]$ are the same as in the main text.

$\tilde{G}_{ij}^{(z)}(\omega)$ have nontrivial dependence under the change of the spatial origin by $\mathbf{d} = (d_x, d_y, d_z)$ because

$$\delta\theta^{(z)}(\omega) = -\sigma_{xy}^H(\omega)d_z,$$

$$\delta\left(-\frac{i}{2}\omega\Gamma_{ijz}(\omega)\right) = -\sigma_{ij}^{\text{sym}}(\omega)d_z,$$

$$\delta\left(-\frac{i}{2}\omega a_{kzz}(\omega)\right) = -\sigma_{kz}^{\text{sym}}(\omega)d_z,$$

(29)

where $\tilde{\sigma}_{ij}(\omega) = -i\omega\tilde{\chi}_{ij}(\omega)$ is the complex-valued optical conductivity tensor, and $\sigma_{ij}^H(\omega) = [\tilde{\sigma}_{ij}(\omega) - \tilde{\sigma}_{ji}(\omega)]/2 = -\omega\chi'_{ij}(\omega)$ is its antisymmetric part.

The origin dependence shows the ambiguity of defining the surface degrees of freedom. Let us recall that we define the surface current density $j^s = \int_{-d_s/2}^{d_s/2} dz j^i(z)$ over the region $-d_s/2 \le z \le d_s/2$. While we can define a physically meaningful value of $d_s$ based on the surface property of a system, this value is still not completely uniquely defined (e.g., if $d_s$ is the surface thickness, $1.01d_s$ also makes sense as a surface thickness). Because of the ambiguity of $d_s$, the amount of the bulk-conductivity contribution to $\tilde{j}^s$ can also vary. For example, let us consider the $xy$ component of the surface conductivity. It has a bulk-conductivity term as well as the magnetic dipole and electric quadrupole contributions. Let us suppose that the surface is defined as the interface between medium 1 (at $z < 0$) and medium 2 at ($z > 0$).

$$\sigma_{xy}^s(\omega) = \tilde{\mathfrak{G}}_{yy}^{(z)}(-d_s/2) - \tilde{\mathfrak{G}}_{yy}^{(z)}(d_s/2) + \int_{-d_s/2}^{d_s/2} dz\sigma_{xy}(\omega,z)$$

$$= \tilde{\mathfrak{G}}_{yy}^{(z)}(-d_s/2) - \tilde{\mathfrak{G}}_{yy}^{(z)}(d_s/2) + \sigma_{xy,1}(\omega)\frac{d_s}{2} + \sigma_{xy,2}(\omega)\frac{d_s}{2}$$

(30)

$$= \tilde{\mathfrak{G}}_{yy,1}^{(z)} + \delta\tilde{\mathfrak{G}}_{yy,1}^{(z)} - \tilde{\mathfrak{G}}_{yy,2}^{(z)} - \delta\tilde{\mathfrak{G}}_{yy,2}^{(z)},$$

where

$$\delta\tilde{\mathfrak{G}}_{yy}^{(z)} = -\sigma_{xy}(\omega)d_z$$

(31)

is the change of $\tilde{\mathfrak{G}}_{yy}^{(z)}$ by the shifting of the origin by $\mathbf{d} = (0, 0, d_z)$. This shows that, while we can define $\sigma_{xy}^s(\omega)$ as the difference of $\tilde{\mathfrak{G}}_{yy}^{(z)}$ between two media for any value of $d_s$, we have to shift the spatial origin by $-d_s/2$ and $d_s/2$ for medium 1 and medium 2, respectively.

## Quantum mechanical expressions of the magneto-electric coupling

**Linear response theory.** The susceptibility tensor for the linear response of an operator $\hat{A}$ to the external field $F_B$

$$A(t) = \left\langle \hat{A}(t)\right\rangle_{F_B=0} + \int dt'\tilde{\chi}_{AB}(t - t')F_B(t')$$

(32)

is given by

$$\tilde{\chi}_{AB}(t - t') = -\frac{i}{\hbar}\Theta(t - t')\langle[\hat{A}(t),\hat{B}(t')]\rangle,$$

(33)

where $\hat{B} = \partial\hat{H}(F)/\partial F_B|_{F_B=0}$ is conjugate to the external field. Here, we take the Heisenberg picture $\hat{O}(t) = e^{iHt}\hat{O}e^{-iHt}$.

In the frequency domain, the susceptibility tensor can be written as

$$\tilde{\chi}_{AB}(\omega) = \int_{-\infty}^{\infty} dt'e^{i\omega(t-t')}\tilde{\chi}_{AB}(t - t')$$

$$= -\frac{1}{\hbar}\sum_{n,m} f_{nm}\frac{\omega_{mn}}{\omega}\frac{\langle n|\hat{A}|m\rangle\langle m|\hat{B}|n\rangle}{\omega_{mn} - \omega} + (\text{FS terms}),$$

(34)

where $|n\rangle$ is the energy eigenstate of the unperturbed Hamiltonian with energy $E_n = \hbar\omega_n$, $\omega_{mn} = \omega_m - \omega_n$, and $f_{nm} = f_n - f_m$ is the difference between the Fermi-Dirac distribution function $f$ of the $|n\rangle$ and $|m\rangle$ states, and the FS terms originate from the Fermi surface, i.e., they have momentum-space derivatives acting on the Fermi-Dirac distribution function ($\partial_{\mathbf{k}}f_n$) in the momentum space representation. The derivation goes as follows. At zero temperature, we have

$$\tilde{\chi}_{AB}(\omega + i\Gamma) = \int_{-\infty}^{\infty} dt'e^{i(\omega+i\Gamma)(t-t')}\tilde{\chi}_{AB}(t - t')$$

$$= -\frac{i}{\hbar}\int_{-\infty}^{\infty} dt'e^{i\omega(t-t')}\Theta(t - t')\langle[\hat{A}(t),\hat{B}(t')]\rangle$$

$$= -\frac{i}{\hbar}\sum_{n,m}\int_{-\infty}^{t} dt'e^{i(\omega+i\Gamma)(t-t')}f_n(\langle n|\hat{A}(t)|m\rangle\langle m|\hat{B}(t')|n\rangle - \langle n|\hat{B}(t')|m\rangle\langle m|\hat{A}(t)|n\rangle)$$

$$= -\frac{i}{\hbar}\sum_{n,m}\int_{-\infty}^{t} dt'e^{i(\omega+i\Gamma)(t-t')}f_n(e^{-i\omega_{mn}(t-t')}\langle n|\hat{A}|m\rangle\langle m|\hat{B}|n\rangle - e^{i\omega_{mn}(t-t')}\langle n|\hat{B}|m\rangle\langle m|\hat{A}|n\rangle)$$

$$= -\frac{i}{\hbar}\sum_{n,m} f_n\left(\frac{1}{-i(\omega+i\Gamma-\omega_{mn})}\langle n|\hat{A}|m\rangle\langle m|\hat{B}|n\rangle - \frac{1}{-i(\omega+i\Gamma+\omega_{mn})}\langle n|\hat{B}|m\rangle\langle m|\hat{A}|n\rangle\right)$$

$$= -\frac{1}{\hbar}\sum_{n,m}\frac{f_{nm}\langle n|\hat{A}|m\rangle\langle m|\hat{B}|n\rangle}{\omega_{mn} - (\omega+i\Gamma)} - \frac{1}{\hbar}\sum_{n,m}\left(f_n\frac{\langle n|\hat{B}|m\rangle\langle m|\hat{A}|n\rangle}{\omega+i\Gamma+\omega_{mn}} - f_m\frac{\langle n|\hat{A}|m\rangle\langle m|\hat{B}|n\rangle}{\omega+i\Gamma+\omega_{nm}}\right)$$

$$= -\frac{1}{\hbar}\sum_{n,m}\frac{f_{nm}\langle n|\hat{A}|m\rangle\langle m|\hat{B}|n\rangle}{\omega_{mn} - (\omega+i\Gamma)} - \frac{1}{\hbar}\frac{1}{\omega+i\Gamma}\sum_n f_n\langle n|[\hat{B}_n,\hat{A}_n]|n\rangle$$

$$= -\frac{1}{\hbar}\sum_{n,m} f_{nm}\frac{\omega_{mn}}{\omega}\frac{\langle n|\hat{A}|m\rangle\langle m|\hat{B}|n\rangle}{\omega_{mn} - (\omega+i\Gamma)}.$$

(35)

where we introduce a finite relaxation rate $\Gamma$ for convergence of time integral, $\hat{O}_n = \hat{P}_n\hat{O}\hat{P}_n$ with $\hat{P}_n = |n\rangle\langle n|$ is the projection of $\hat{O}$ to states $|n\rangle$, and $\sum_n f_n\langle n|[\hat{B}_n,\hat{A}_n]|n\rangle = \sum_{n\neq m} f_{nm}A_{nm}B_{mn}$.

It is often convenient to separate the real and imaginary parts of $\langle n|\hat{A}|m\rangle\langle m|\hat{B}|n\rangle$ as follows.

$$\tilde{\chi}_{AB}(\omega) = -\frac{1}{2\hbar}\sum_{n,m} f_{nm}\frac{1}{\omega}\left[\frac{\omega_{mn}}{\omega_{mn} - \omega} - \frac{\omega_{nm}}{\omega_{nm} - \omega}\right]\text{Re}\left[\langle n|\hat{A}|m\rangle\langle m|\hat{B}|n\rangle\right]$$

$$- \frac{i}{2\hbar}\sum_{n,m} f_{nm}\frac{1}{\omega}\left[\frac{\omega_{mn}}{\omega_{mn} - \omega} + \frac{\omega_{nm}}{\omega_{nm} - \omega}\right]\text{Im}\left[\langle n|\hat{A}|m\rangle\langle m|\hat{B}|n\rangle\right]$$

$$= -\frac{1}{\hbar}\sum_{n,m} f_{nm}\left(\frac{\omega_{mn}}{\omega_{mn}^2 - \omega^2}\text{Re}\left[\langle n|\hat{A}|m\rangle\langle m|\hat{B}|n\rangle\right] + i\frac{\omega_{mn}^2/\omega}{\omega_{mn}^2 - \omega^2}\text{Im}\left[\langle n|\hat{A}|m\rangle\langle m|\hat{B}|n\rangle\right]\right)$$

$$= \chi_{AB}(\omega) - i\chi'_{AB}(\omega),$$

(36)

where we assume $\hat{A}$ and $\hat{B}$ are Hermitian operators.

Let us take electric susceptibility $\tilde{\chi}_{ij}$ as an example, which is defined by

$$P_i(t) = \left\langle\hat{P}_i\right\rangle_{E=0} + \int dt'\tilde{\chi}_{ij}(t - t')E_j(t').$$

(37)

Then, $\hat{A} = \hat{P}_i$ is the electric polarization density and $\hat{B} = -\hat{P}_jV$ is the polarization, so we have

$$\tilde{\chi}_{ij}(\omega) = \frac{V}{\hbar}\sum_{n,m} f_{nm}\frac{\omega_{mn}}{\omega(\omega_{mn} - \omega)}\langle n|\hat{P}_i|m\rangle\langle m|\hat{P}_j|n\rangle$$

$$= \frac{2V}{\hbar}\sum_{n,m} f_n\frac{\omega_{mn}}{\omega_{mn}^2 - \omega^2}\text{Re}\left[\langle n|\hat{P}_i|m\rangle\langle m|\hat{P}_j|n\rangle\right]$$

$$+ i\frac{2V}{\hbar}\sum_{n,m} f_n\frac{\omega_{mn}^2/\omega}{\omega_{mn}^2 - \omega^2}\text{Im}\left[\langle n|\hat{P}_i|m\rangle\langle m|\hat{P}_j|n\rangle\right]$$

$$= \chi_{ij}(\omega) - i\chi'_{ij}(\omega)$$

(38)

Similarly, other susceptibility tensors are given by

$$\tilde{a}_{ijk}(\omega) = \frac{V}{\hbar}\sum_{n,m} f_{nm}\frac{\omega_{mn}}{\omega(\omega_{mn}-\omega)}\langle n|\hat{P}_i|m\rangle\langle m|\hat{Q}_{jk}|n\rangle = a_{ijk} - ia'_{ijk},$$

$$\tilde{\mathfrak{a}}_{ijk}(\omega) = \frac{V}{\hbar}\sum_{n,m} f_{nm}\frac{\omega_{mn}}{\omega(\omega_{mn}-\omega)}\langle n|\hat{Q}_{ij}|m\rangle\langle m|\hat{P}_k|n\rangle = a_{kij} + ia'_{kij},$$

$$\tilde{G}_{ij}(\omega) = \frac{V}{\hbar}\sum_{n,m} f_{nm}\frac{\omega_{mn}}{\omega(\omega_{mn}-\omega)}\langle n|\hat{P}_i|m\rangle\langle m|\hat{M}_j|n\rangle = G_{ij} - iG'_{ij},$$

$$\breve{\mathfrak{G}}_{ij}(\omega) = \frac{V}{\hbar}\sum_{n,m} f_{nm}\frac{\omega_{mn}}{\omega(\omega_{mn}-\omega)}\langle n|\hat{M}_i|m\rangle\langle m|\hat{P}_j|n\rangle = G_{ji} + iG'_{ji}. \quad (39)$$

**Structure of the optical magneto-electric coupling.** The bare magneto-electric coupling can be decomposed into three parts as follows.

$$G_{ij}(\omega) = G^{K}_{ij}(\omega) + G^{C}_{ij}(\omega) + G^{A}_{ij}(\omega)$$
$$= \frac{e^2}{\hbar V}\epsilon_{jkl}\sum_{n\in occ,m\in unocc,\mathbf{k}}\frac{\omega_{mn}}{\omega_{mn}^2-\omega^2}\mathrm{Re}\left[\sum_{n'} r^i_{nm}r^k_{mn'}v^l_{n'n} + \sum_{m'} r^i_{nm}v^l_{mm'}r^k_{m'n}\right]$$
$$+ \frac{e^2}{\hbar V}\epsilon_{jkl}\sum_{n,n'\in occ,m\in unocc,\mathbf{k}}\frac{\omega_{mn}^2}{\omega_{mn}^2-\omega^2}\mathrm{Im}\left[r^i_{nm}r^k_{mn'}r^l_{n'n}\right]$$
$$+ \frac{e^2}{\hbar V}\epsilon_{jkl}\sum_{n\in occ,m\in unocc,\mathbf{k}}\frac{\omega^2\omega_{mn}}{(\omega_{mn}^2-\omega^2)^2}\mathrm{Re}\left[r^i_{nm}r^k_{mn}\right](v^l_n + v^l_m). \quad (40)$$

The $K$-term and $C$-term correspond to the Kubo-like and Chern-Simons terms in ref. 19 of the static limit. The last $A$-term is nonzero only when $\omega \neq 0$. While the $C$-term was called the Chern-Simons term, it has additional terms, actually. In the static limit,

$$G^{C}_{ij}(0) = \delta_{ij}\frac{\theta_{CS}e^2}{2\pi h} + \frac{1}{3}\epsilon_{jkl}\lim_{\omega\to 0}\left[\omega a'_{kli}(\omega)\right] + \frac{e^2}{6\hbar}\epsilon_{jkl}\int_{\mathbf{k}}\partial_k g_{il}, \quad (41)$$

Here, $\theta_{CS}$ is the axion angle given by the Chern-Simons integral in the three-dimensional Brillouin zone[17,18]

$$\theta_{CS} = -\frac{4\pi^2}{3V}\epsilon_{ijk}\mathrm{Im}\mathrm{Tr}\left[P\hat{r}^i P\hat{r}^j P\hat{r}^k\right]$$
$$= -\frac{1}{4\pi}\epsilon_{ijk}\int_{BZ}d^3k\left[\left(A^i\frac{1}{2}F^{jk} + \frac{i}{3}A^iA^jA^k\right)\right.$$
$$\left. - \sum_{n,n'\in occ}\partial_j\mathrm{Re}\left(\langle i\partial_i\psi_n|\psi_{n'}\rangle A^k_{n'n}\right)\right], \quad (42)$$

where $P$ is the projection to the occupied states, and $A^k_{n'n} = \langle u_{n'}|i\partial_k|u_n\rangle$ is the non-abelian Berry connection for the occupied states. Since the last term vanishes in insulators with vanishing Chern number, only the Chern-Simons integral remains in the expression. The quadrupole term takes the form $\lim_{\omega\to 0}\left[\omega a'_{kli}(\omega)\right] = -\frac{e^2}{\hbar V}\epsilon_{jkl}\mathrm{Im}\mathrm{Tr}[P\hat{r}^k Q\hat{r}^l\hat{r}^i]$. Lastly, $g_{ij} = \sum_{n\in occ,m\in unocc}r^i_{nm}r^j_{mn}$ is the quantum metric of the occupied states. The quantum metric term in Eq. (41) cancels the Fermi surface contribution from the quadrupole term (the quantum metric contribution in electric quadrupole responses was discussed in[47,48]), such that there is no Fermi surface contribution to the magneto-electric coupling $G_{ij}$. In comparison, note that its time-reversal-symmetric counterpart $G'_{ij}$ has a Fermi surface contribution, which gives rise to natural optical activity in metals, termed gyrotropic magnetic effect[49,50]. The quadrupole and quantum metric terms were missed in previous studies[17,18], but they do not affect the axion angle.

Let us derive Eq. (42). A key equation in our derivation is

$$\sum_{j,k}\epsilon_{ijk}\left\langle i\partial_j\psi_m|\psi_p\right\rangle\langle\psi_p|i\partial_k\psi_n\rangle = \sum_{j,k}\epsilon_{ijk}\langle\partial_j\psi_m|\partial_k\psi_n\rangle = 0, \quad (43)$$

which follows from the Wannier representation:

$$\sum_{j,k}\epsilon_{ijk}\langle\partial_j\psi_m|\partial_k\psi_n\rangle = \epsilon_{ijk}\frac{1}{N}\sum_{j,k,\mathbf{R},\mathbf{R}'}R_jR'_k e^{i(\mathbf{k}\cdot\mathbf{R}-\mathbf{k}'\cdot\mathbf{R}')}\langle w_{m\mathbf{R}'}|w_{n\mathbf{R}}\rangle$$
$$= \delta_{mn}\frac{1}{N}\sum_{\mathbf{R}}e^{i\mathbf{k}\cdot(\mathbf{R}-\mathbf{R}')}\sum_{j,k}\epsilon_{ijk}R_jR_k \quad (44)$$
$$= 0.$$

Using this identity, we can show that

$$\sum_{i,j,k}\epsilon_{ijk}\mathrm{Tr}\mathrm{Im}\left[P\hat{r}^i P\hat{r}^j P\hat{r}^k\right] = -\sum_{i,j,k}\epsilon_{ijk}\mathrm{Tr}\mathrm{Im}\left[P\hat{r}^i P\hat{r}^j Q\hat{r}^k\right]$$
$$= \sum_{i,j,k}\epsilon_{ijk}\mathrm{Re}\left[\left(A^i_{nn'} - \langle i\partial_i\psi_n|\psi_{n'}\rangle\right)\frac{1}{2}F^{jk}_{n'n}\right]$$
$$= \sum_{i,j,k}\epsilon_{ijk}\mathrm{Re}\left[A^i_{nn'}\frac{1}{2}F^{jk}_{n'n} - \langle i\partial_i\psi_n|\psi_{n'}\rangle\left(\partial_j A^k_{n'n} - i(A^jA^k)_{n'n}\right)\right]$$
$$= \sum_{i,j,k}\epsilon_{ijk}\left[A^i_{nn'}\frac{1}{2}F^{jk}_{n'n} + i\langle i\partial_i\psi_n|\psi_{n'}\rangle(A^jA^k)_{n'n} - \partial_j\mathrm{Re}\left(\langle i\partial_i\psi_n|\psi_{n'}\rangle A^k_{n'n}\right)\right]$$
$$= \sum_{i,j,k}\epsilon_{ijk}\left[A^i_{nn'}\frac{1}{2}F^{jk}_{n'n} - i\frac{1}{3}\mathrm{Tr}\left[P\hat{r}^i P\hat{r}^j P\hat{r}^k - A^iA^jA^k\right] - \partial_j\mathrm{Re}\left(\langle i\partial_i\psi_n|\psi_{n'}\rangle A^k_{n'n}\right)\right]$$
$$= \frac{1}{3}\sum_{i,j,k}\epsilon_{ijk}\mathrm{Tr}\mathrm{Im}\left[P\hat{r}^i P\hat{r}^j P\hat{r}^k\right] + \sum_{i,j,k}\epsilon_{ijk}\left[\mathrm{Tr}\left(A^i\frac{1}{2}F^{jk} + \frac{i}{3}A^iA^jA^k\right)\right.$$
$$\left. - \partial_j\mathrm{Re}\left(\langle i\partial_i\psi_n|\psi_{n'}\rangle A^k_{n'n}\right)\right]. \quad (45)$$

It follows that

$$\frac{1}{3}\mathrm{Tr}G = \frac{e^2}{\hbar}\frac{1}{3}\epsilon_{ijk}\mathrm{Tr}\mathrm{Im}\left[P\hat{r}^i P\hat{r}^j P\hat{r}^k\right]$$
$$= \frac{e^2}{\hbar}\frac{1}{2}\epsilon_{ijk}\left[\mathrm{Tr}\left(A^i\frac{1}{2}F^{jk} + \frac{i}{3}A^iA^jA^k\right) - \partial_j\mathrm{Re}\left(\langle i\partial_i\psi_n|\psi_{n'}\rangle A^k_{n'n}\right)\right]. \quad (46)$$

This is equivalent to Eq. (42). At finite $\omega$, the trace part of the magneto-electric coupling is not represented as a Chern-Simons integral by the same approach.

**Gauge-invariant expressions.** Since $\tilde{G}^{(z)}_{ij}(\omega)$ and $\tilde{G}^{(z,2)}_{ij}(\omega)$ are origin dependent only along the $z$ direction, they can be calculated gauge independently when the $z$ direction has open boundaries. Here we derive such gauge-invariant expressions. We focus on the orbital magneto-electric coupling to simplify expressions. It is straightforward to include spin parts as the spin magnetic moment is well defined in momentum space. The expressions of $T_{ij}$, $T'_{ij}$ and $S_{ijk}$ are also given for completeness.

Let us begin with $\tilde{G}^{(z)}_{ix}(\omega)$.

$$\tilde{G}^{(z)}_{ix}(\omega) = \tilde{G}_{ix}(\omega) - \frac{i}{2}\omega\tilde{a}_{iyz}(\omega)$$
$$= \frac{V}{\hbar}\sum_{n\neq m} f_{nm}\frac{\langle n|\hat{P}_i|m\rangle\langle m|\hat{m}_x|n\rangle - \frac{i}{2}\omega_{mn}\langle n|\hat{P}_i|m\rangle\langle m|\hat{Q}_{yz}|n\rangle}{\omega_{mn}-\omega}$$
$$= \frac{e^2}{\hbar V}\sum_{n\neq m}\frac{f_{nm}}{\omega_{mn}-\omega}\left(r^i_{nm}\langle m|\frac{1}{2}(\hat{r}^y\hat{v}^z - \hat{r}^z\hat{v}^y)|n\rangle - \frac{i}{2}\omega_{mn}r^i_{nm}\langle m|\hat{r}^y\hat{r}^z|n\rangle\right) \quad (47)$$
$$= \frac{e^2}{\hbar V}\sum_{n\neq m}\frac{f_{nm}}{\omega_{mn}-\omega}r^i_{nm}\langle m|\frac{1}{2}\left(\hat{r}^y\hat{v}^z - \hat{r}^z\hat{v}^y - \frac{1}{i\hbar}[\hat{r}^y\hat{r}^z,\hat{H}]\right)|n\rangle$$
$$= \frac{e^2}{\hbar V}\sum_{n\neq m}\frac{f_{nm}}{\omega_{mn}-\omega}r^i_{nm}\langle m|\frac{1}{2}(-\hat{v}^y\hat{r}^z - \hat{r}^z\hat{v}^y)|n\rangle.$$

Similarly,

$$\tilde{G}^{(z)}_{iy}(\omega) = \tilde{G}_{iy}(\omega) + \frac{i}{2}\omega\tilde{a}_{ixz}(\omega) = \frac{e^2}{\hbar V}\sum_{n\neq m}\frac{f_{nm}}{\omega_{mn}-\omega}r^i_{nm}\langle m|\frac{1}{2}(\hat{v}^x\hat{r}^z + \hat{r}^z\hat{v}^x)|n\rangle. \quad (48)$$

For the $zz$ component, we obtain

$$\tilde{G}^{(z)}_{zz}(\omega) = \tilde{G}_{zz} - \frac{i}{2}\omega\left[\tilde{a}_{zxy}(\omega) - \tilde{a}_{zyx}(\omega)\right]$$
$$= \frac{e^2}{2\hbar V}\sum_{n\neq m}\frac{f_{nm}}{\omega_{mn}-\omega}\sum_{p:E_p\neq E_m}\left[r^z_{nm}r^x_{mp}v^y_{pn} - r^z_{np}r^x_{pm}v^y_{mn} - (x\leftrightarrow y)\right] \quad (49)$$

by using

$$
\begin{aligned}
\bar{G}_{zz} &= \frac{V}{\hbar}\sum_{n\ne m} f_{nm} \frac{\langle n|\hat{P}_z|m\rangle\langle m|\hat{M}_z|n\rangle}{\omega_{mn}-\omega}\\
&= \frac{e^2}{2\hbar V}\sum_{n\ne m}\frac{f_{nm}}{\omega_{mn}-\omega}r_{nm}^z\left[\sum_{p\ne m}(r_{mp}^x v_{pn}^y - r_{mp}^y v_{pn}^x) + i\omega_{mn}(r_{mm}^x r_{mn}^y - r_{mm}^y r_{mn}^x)\right]\\
&= \frac{e^2}{2\hbar V}\sum_{n\ne m}\frac{f_{nm}}{\omega_{mn}-\omega}\left[r_{nm}^z\sum_{p\ne m}(r_{mp}^x v_{pn}^y - r_{mp}^y v_{pn}^x) + i\omega_{mn}\left(\left[(\hat{r}^z\hat{r}^x)_{nm} - \sum_{p\ne m}r_{np}^z r_{pm}^x\right]r_{mn}^y - (x\to y)\right)\right]\\
&= \frac{e^2}{2\hbar V}\sum_{n\ne m}\frac{f_{nm}}{\omega_{mn}-\omega}\left[r_{nm}^z\sum_{p:E_p\ne E_m}(r_{mp}^x v_{pn}^y - r_{mp}^y v_{pn}^x) - i\omega_{mn}\sum_{p:E_p\ne E_m}\left[r_{np}^z r_{pm}^x - (x\leftrightarrow y)\right]r_{mn}^y\right]\\
&\quad + \frac{i}{2}\omega\left[\bar{a}_{zxy}(\omega) - \bar{a}_{zyx}(\omega)\right].
\end{aligned}
$$

(50)

For the $zx$ component, we can follow the strategy for the $zz$ component

$$
\begin{aligned}
\bar{G}_{zx} &= \frac{V}{\hbar}\sum_{n\ne m} f_{nm}\frac{\langle n|\hat{P}_z|m\rangle\langle m|\hat{M}_x|n\rangle}{\omega_{mn}-\omega}\\
&= \frac{e^2}{2\hbar V}\sum_{n\ne m}\frac{f_{nm}}{\omega_{mn}-\omega}r_{nm}^z\left[\sum_{p\ne m}r_{mp}^y v_{pn}^z - \sum_p r_{mp}^z v_{pn}^y + i\omega_{mn}r_{mm}^y r_{mn}^z\right]\\
&= \frac{e^2}{2\hbar V}\sum_{n\ne m}\frac{f_{nm}}{\omega_{mn}-\omega}\left[r_{nm}^z\sum_{p\ne m}r_{mp}^y v_{pn}^z - r_{nm}^z\sum_p r_{mp}^z v_{pn}^y + i\omega_{mn}\left(\left[(\hat{r}^z\hat{r}^y)_{nm} - \sum_{p\ne m}r_{np}^z r_{pm}^y\right]r_{mn}^z\right)\right]\\
&= \frac{e^2}{2\hbar V}\sum_{n\ne m}\frac{f_{nm}}{\omega_{mn}-\omega}\left[r_{nm}^z\sum_{p:E_p\ne E_m}r_{mp}^y v_{pn}^z - r_{nm}^z\sum_p r_{mp}^z v_{pn}^y - i\omega_{mn}\sum_{p:E_p\ne E_m}r_{np}^z r_{pm}^y r_{mn}^z\right]\\
&\quad + \frac{i}{2}\omega\bar{a}_{zyz}(\omega)
\end{aligned}
$$

(51)

to define

$$
\begin{aligned}
\bar{G}_{zx}^{(z,2)} &= \bar{G}_{zx} - \frac{i}{2}\omega\bar{a}_{yzz}(\omega)\\
&= \frac{e^2}{2\hbar V}\sum_{n\ne m}\frac{f_{nm}}{\omega_{mn}-\omega}\left[r_{nm}^z\left(\sum_{p:E_p\ne E_m}r_{mp}^y v_{pn}^z - \sum_p r_{mp}^z v_{pn}^y\right) - \sum_{p:E_p\ne E_m}r_{np}^z r_{pm}^y v_{mn}^z\right].
\end{aligned}
$$

(52)

Similarly, we define

$$
\begin{aligned}
\bar{G}_{zy}^{(z,2)} &= \bar{G}_{zy} + \frac{i}{2}\omega\bar{a}_{xzz}(\omega)\\
&= -\frac{e^2}{2\hbar V}\sum_{n\ne m}\frac{f_{nm}}{\omega_{mn}-\omega}\left[r_{nm}^z\left(\sum_{p:E_p\ne E_m}r_{mp}^x v_{pn}^z - \sum_p r_{mp}^z v_{pn}^x\right) - \sum_{p:E_p\ne E_m}r_{np}^z r_{pm}^x v_{mn}^z\right].
\end{aligned}
$$

(53)

Note that $G_{zi}^{(z)} = G_{zi}^{(z,2)}$ while $G_{zi}'^{(z)} \ne G_{zi}'^{(z,2)}$ for $i = x, y$.

Fully origin-independent bulk response functions $T_{ij}$, $T_{ij}'$ and $S_{ijk}$ can be calculated from the bulk conductivity tensor defined by $J_i = \sum_{j,k}\sigma_{ijk}q_j E_k$[19]:

$$
\begin{aligned}
T_{ij}(\omega) &= \frac{1}{3i}\epsilon_{jkl}\sigma_{ikl},\\
T_{ij}'(\omega) &= \frac{1}{8i}\epsilon_{jkl}\left[2(\sigma_{ikl} - \sigma_{kil}) - (\sigma_{kli} - \sigma_{lki})\right],\\
S_{ijk}(\omega) &= -\frac{1}{6i}\left(\sigma_{ijk} + \sigma_{jki} + \sigma_{kij} + \sigma_{jik} + \sigma_{ikj} + \sigma_{kji}\right),
\end{aligned}
$$

(54)

where

$$
\begin{aligned}
\sigma_{ikl} &= \frac{ie^2}{\hbar\omega}\sum_{n,m}\int_{\mathbf{k}} f_{nm}\left[\frac{\omega_{mn}}{\omega_{mn}-\omega}\left((r_{nm}^j B_{mn}^{ik})^* + r_{nm}^i B_{mn}^{jk}\right) + \frac{\omega_{mn}^2}{2(\omega_{mn}-\omega)^2}r_{nm}^i r_{mn}^j(v_{mm}^k + v_{nn}^k)\right]\\
&\quad - \frac{ie^2}{\hbar\omega^2}\sum_{n,m}\int_{\mathbf{k}}\partial_k f_n v_{nn}^i v_{nn}^j
\end{aligned}
$$

(55)

and

$$
B_{mn}^{kl} = \frac{1}{2}\left(\sum_{p\ne m}r_{mp}^k v_{pn}^l + \sum_{p\ne n}v_{mp}^l r_{pn}^k\right).
$$

(56)

## Decomposition of the position operator

Here we derive Eq. (9). Let us consider the matrix element of the position operator in the Bloch state basis. We transform it to the Wannier basis $|w_{\alpha\mathbf{R}}\rangle$ by

$$
\begin{aligned}
\langle\psi_{m\mathbf{k}'}|\hat{r}^z|\psi_{n\mathbf{k}}\rangle &= \sum_{\beta,\mathbf{R}';\alpha,\mathbf{R}}\langle\psi_{m\mathbf{k}'}|w_{\beta\mathbf{R}'}\rangle\langle w_{\beta\mathbf{R}'}|\hat{r}^z|w_{\alpha\mathbf{R}}\rangle\langle w_{\alpha\mathbf{R}}|\psi_{n\mathbf{k}}\rangle\\
&= \sum_{\beta,\mathbf{R}';\alpha,\mathbf{R}}\langle\psi_{m\mathbf{k}'}|w_{\beta\mathbf{R}'}\rangle\langle w_{\beta 0}|\hat{T}_{\mathbf{R}'}^\dagger\hat{r}^z\hat{T}_{\mathbf{R}}|w_{\alpha\mathbf{R}-\mathbf{R}'}\rangle\langle w_{\alpha\mathbf{R}}|\psi_{n\mathbf{k}}\rangle\\
&= \sum_{\beta,\mathbf{R}';\alpha,\mathbf{R}}\langle\psi_{m\mathbf{k}'}|w_{\beta\mathbf{R}'}\rangle(R^z\delta_{\alpha\beta}\delta_{\mathbf{R},\mathbf{R}'} + \langle w_{\beta 0}|\hat{r}^z|w_{\alpha(\mathbf{R}-\mathbf{R}')}\rangle)\langle w_{\alpha\mathbf{R}}|\psi_{n\mathbf{k}}\rangle\\
&= \sum_{\beta,\mathbf{R}';\alpha,\mathbf{R}}\langle\psi_{m\mathbf{k}'}|w_{\beta\mathbf{R}'}\rangle\langle w_{\beta 0}|\hat{r}^z|w_{\alpha(\mathbf{R}-\mathbf{R}')}\rangle\langle w_{\alpha\mathbf{R}}|\psi_{n\mathbf{k}}\rangle + \sum_{\alpha,\mathbf{R}}\langle\psi_{m\mathbf{k}'}|w_{\alpha\mathbf{R}}\rangle R^z\langle w_{\alpha\mathbf{R}}|\psi_{n\mathbf{k}}\rangle,
\end{aligned}
$$

(57)

where we use $|w_{\alpha\mathbf{R}}\rangle = \hat{T}_{\mathbf{R}}|w_{\alpha 0}\rangle$ in the second line, where $\hat{T}_{\mathbf{R}}$ is the translation operator by $\mathbf{R}$, and use $\hat{T}_{\mathbf{R}'}^\dagger\hat{r}^z\hat{T}_{\mathbf{R}'} = \hat{r}^z + \mathbf{R}'^z$ and the ortho-normality of the Wannier states $\langle w_{\beta\mathbf{R}'}|w_{\alpha\mathbf{R}}\rangle = \delta_{\alpha\beta}\delta_{\mathbf{R},\mathbf{R}'}$ in the third line. To get Eq. (9) from Eq. (57), we define $\mathbb{A}_{\beta\alpha}^z(\mathbf{k}) = \sum_{\mathbf{R}}\langle w_{\beta 0}|\hat{r}^z|w_{\alpha\mathbf{R}}\rangle e^{i\mathbf{k}\cdot\mathbf{R}}$, such that

$$
\langle w_{\beta 0}|\hat{r}^z|w_{\alpha\mathbf{R}}\rangle = \frac{1}{N}\sum_{\mathbf{k}}\mathbb{A}_{\beta\alpha}^z(\mathbf{k})e^{-i\mathbf{k}\cdot\mathbf{R}},
$$

(58)

where $N$ is the number of $\mathbf{k}$ points. Then, the first term in the last line of Eq. (57) becomes

$$
\begin{aligned}
&\sum_{\beta,\mathbf{R}';\alpha,\mathbf{R}}\langle\psi_{m\mathbf{k}'}|w_{\beta\mathbf{R}'}\rangle\langle w_{\beta 0}|\hat{r}^z|w_{\alpha(\mathbf{R}-\mathbf{R}')}\rangle\langle w_{\alpha\mathbf{R}}|\psi_{n\mathbf{k}}\rangle\\
&= \sum_{\beta,\mathbf{R}';\alpha,\mathbf{R}}\langle\psi_{m\mathbf{k}'}|w_{\beta 0}\rangle\frac{1}{N}\sum_{\mathbf{k}''}\mathbb{A}_{\beta\alpha}^z(\mathbf{k}'')e^{-i\mathbf{k}''\cdot(\mathbf{R}-\mathbf{R}')}e^{i(\mathbf{k}\cdot\mathbf{R}-\mathbf{k}'\cdot\mathbf{R}')}\langle w_{\alpha 0}|\psi_{n\mathbf{k}}\rangle\\
&= \sum_{\alpha,\beta,\mathbf{R}'}\langle\psi_{m\mathbf{k}'}|w_{\beta 0}\rangle\mathbb{A}_{\beta\alpha}^z(\mathbf{k})e^{i\mathbf{k}\cdot\mathbf{R}'}e^{-i\mathbf{k}'\cdot\mathbf{R}'}\langle w_{\alpha 0}|\psi_{n\mathbf{k}}\rangle\\
&= \delta_{\mathbf{k},\mathbf{k}'}N\sum_{\alpha,\beta}\langle\psi_{m\mathbf{k}}|w_{\beta 0}\rangle\mathbb{A}_{\beta\alpha}^z(\mathbf{k})\langle w_{\alpha 0}|\psi_{n\mathbf{k}}\rangle\\
&= \delta_{\mathbf{k},\mathbf{k}'}\sum_{\alpha,\beta}\langle\psi_{m\mathbf{k}}|\psi_{\beta\mathbf{k}}\rangle\mathbb{A}_{\beta\alpha}^z(\mathbf{k})\langle\psi_{\alpha\mathbf{k}}|\psi_{n\mathbf{k}}\rangle,
\end{aligned}
$$

(59)

and we arrive at Eq. (9).

$$
\begin{aligned}
\langle\psi_{m\mathbf{k}'}|\hat{r}^z|\psi_{n\mathbf{k}}\rangle &= \delta_{\mathbf{k},\mathbf{k}'}\sum_{\beta,\alpha}\langle\psi_{m\mathbf{k}}|\psi_{\beta\mathbf{k}}\rangle\mathbb{A}_{\beta\alpha}^z(\mathbf{k})\langle\psi_{\alpha\mathbf{k}}|\psi_{n\mathbf{k}}\rangle\\
&\quad + \sum_{\alpha,\mathbf{R}}\langle\psi_{m\mathbf{k}'}|w_{\alpha\mathbf{R}}\rangle R^z\langle w_{\alpha\mathbf{R}}|\psi_{n\mathbf{k}}\rangle.
\end{aligned}
$$

(60)

The second term is nonzero only when $n = m$ because

$$
\begin{aligned}
\sum_{\alpha,\mathbf{R}}\langle\psi_{m\mathbf{k}'}|w_{\alpha\mathbf{R}}\rangle R^z\langle w_{\alpha\mathbf{R}}|\psi_{n\mathbf{k}}\rangle &= \frac{1}{N}\sum_{\alpha,\mathbf{R},\mathbf{k}_1,\mathbf{k}_2}e^{i(\mathbf{k}_2-\mathbf{k}_1)\cdot\mathbf{R}}\langle\psi_{m\mathbf{k}'}|\psi_{\alpha\mathbf{k}_1}\rangle R^z\langle\psi_{\alpha\mathbf{k}_2}|\psi_{n\mathbf{k}}\rangle\\
&= \frac{1}{N}\sum_{\mathbf{R}}e^{i(\mathbf{k}-\mathbf{k}')\cdot\mathbf{R}}R^z\sum_\alpha\langle\psi_{m\mathbf{k}'}|\psi_{\alpha\mathbf{k}'}\rangle\langle\psi_{\alpha\mathbf{k}}|\psi_{n\mathbf{k}}\rangle\\
&= -i\partial_{k^z}\delta_{\mathbf{k},\mathbf{k}'}\sum_\alpha\langle\psi_{m\mathbf{k}'}|\psi_{\alpha\mathbf{k}'}\rangle\langle\psi_{\alpha\mathbf{k}}|\psi_{n\mathbf{k}}\rangle\\
&= -i\delta_{mn}\partial_{k^z}\delta_{\mathbf{k},\mathbf{k}'},
\end{aligned}
$$

(61)

where we use that $\partial_{k^z}\delta_{\mathbf{k},\mathbf{k}'} \ne 0$ requires $\mathbf{k}' \to \mathbf{k}$, and $\sum_\alpha\langle\psi_{m\mathbf{k}}|\psi_{\alpha\mathbf{k}}\rangle\langle\psi_{\alpha\mathbf{k}}|\psi_{n\mathbf{k}}\rangle = \delta_{mn}$.

## Macroscopic electrodynamics in the medium

Electric displacement $\mathbf{D}$ and magnetic field $\mathbf{H}$ satisfying Maxwell's equations

$$
\begin{aligned}
\nabla\cdot\mathbf{D} &= \rho^f,\\
\nabla\times\mathbf{H} &= \mathbf{J}^f + \dot{\mathbf{D}}
\end{aligned}
$$

(62)

are defined by

$$\begin{pmatrix} \mathbf{D} \\ \mathbf{H} \end{pmatrix} = \begin{pmatrix} \tilde{A} & \tilde{T} \\ \tilde{U} & \tilde{X} \end{pmatrix} \begin{pmatrix} \mathbf{E} \\ \mathbf{B} \end{pmatrix}, \tag{63}$$

where $\tilde{F} = F - iF'$ for $F = A, T, U, X$, and

$$
\begin{aligned}
A_{ij} &= \epsilon_0 \delta_{ij} + \chi_{ij} + \frac{1}{3}(a'_{ijk} + a'_{jki} + a'_{kij})k_k, \\
A'_{ij} &= \chi'_{ij}, \\
T_{ij} &= G_{ij} - \frac{1}{3}\delta_{ij}G_{kk} - \frac{1}{6}\omega\epsilon_{jkl}a'_{kli}, \\
T'_{ij} &= G'_{ij} - \frac{1}{2}\omega\epsilon_{jkl}a_{kli}, \\
U_{ij} &= -U_{ji}, \\
U'_{ij} &= U_{ji}, \\
X_{ij} &= \mu_0^{-1}\delta_{ij}, \\
X'_{ij} &= 0
\end{aligned} \tag{64}
$$

up to electric quadrupole-magnetic dipole order. While Maxwell's equations do not uniquely specify the form of **D** and **H**, additional requirements from the reciprocity relations and spatial-origin independence gives Eq. (64) as shown in[25]. Here, the free charge and current $\rho^f$ and $\mathbf{J}^f$ are boundary charge and current appearing due to the change of material properties across the interface that is not described by $\tilde{T}_{ij}$, $\tilde{U}_{ij}$, and $S_{ijk} = (a'_{ijk} + a'_{jki} + a'_{kij})/3$.

In $PT$-symmetric systems where $G'_{ij} = a_{ijk} = 0$,

$$
\begin{aligned}
\rho^f &= -B_j\partial_i\left(G_{ij} - T_{ij} - \frac{1}{2}\epsilon_{jkl}a'_{kli}\right) + \frac{i}{2}(\partial_j E_k + \partial_k E_j)\partial_i(a'_{kij} - S_{kij}) \\
&\quad + \frac{i}{2}E_k\partial_i\partial_j(a'_{kij} - S_{kij}),
\end{aligned} \tag{65}
$$

$$
J_i^f = E_j\partial_k\left[\epsilon_{ikl}(G_{jl} - T_{ji}) - \frac{\omega}{2}(a'_{jik} - S_{jik})\right],
$$

**Wave equation.** The wave equation up to electric quadrupole/magnetic dipole takes the following form[21]:

$$\left[\delta_{ij} + \epsilon_0^{-1}\tilde{\chi}_{ij} + i\mu_0 nc\sum_k \kappa_k\tilde{\sigma}_{ijk} + n^2(\kappa_i\kappa_j - \delta_{ij})\right]E_j = 0 \tag{66}$$

where $\tilde{\chi}_{ij} = \chi_{ij} - i\chi'_{ij}$ satisfying $\chi_{ij} = \chi_{ji}$ and $\chi'_{ij} = -\chi'_{ji}$, $\kappa_i = k_i/|\mathbf{k}|$ is the propagation direction of light, $n$ is the refractive index, $\tilde{\sigma}_{ijk} = \sigma_{ijk} - i\sigma'_{ijk}$ is the complex bulk conductivity coefficient defined by $\tilde{\sigma}_{ij}(\mathbf{q}) = \tilde{\sigma}_{ij}(0) + \tilde{\sigma}_{ijk}q_k + \dots$, and

$$
\begin{aligned}
\sigma_{ijk} &= i\left[\epsilon_{ikl}G_{jl} + \epsilon_{jkl}G_{il} - \frac{1}{2}\omega(a'_{ijk} + a'_{jik})\right] = \sigma_{jik}, \\
\sigma'_{ijk} &= i\left[-\epsilon_{ikl}G'_{jl} + \epsilon_{jkl}G'_{il} + \frac{1}{2}\omega(a_{ijk} - a_{jik})\right] = -\sigma'_{jik}.
\end{aligned} \tag{67}
$$

Let us assume $C_{3z}$ and $C_{2x}$ symmetries for simplicity. We further impose that the bulk Hall response is zero, i.e., $\chi'_{ij} = 0$, in order to focus on the magneto-electric and electric-quadrupole effects. For $\boldsymbol{\kappa} = \pm\hat{z}$, the wave equation is

$$\begin{pmatrix} 1 + \epsilon_0^{-1}\chi_{xx} - n^2 & n\kappa_z\mu_0 c\sigma'_{xyz} \\ -n\kappa_z\mu_0 c\sigma'_{xyz} & 1 + \epsilon_0^{-1}\chi_{xx} - n^2 \end{pmatrix} \begin{pmatrix} E_x \\ E_y \end{pmatrix} = 0. \tag{68}$$

The refractive index satisfying the wave equation is given by

$$n_\pm^{\kappa_z} = \sqrt{1 + \epsilon_0^{-1}\chi_{xx} + (i\mu_0 c\sigma'_{xyz}/2)^2} \mp i\kappa_z\mu_0 c\sigma'_{xyz}/2, \tag{69}$$

for circular polarization $\hat{\pm} = \hat{x} + i\hat{y}$.

**Reflection and transmission from a single interface.** We consider the interface of medium 1 ($z > 0$) and medium 2 ($z < 0$) with the surface normal $\hat{z}$. For normal incidence, in the circularly polarized basis,

$$\begin{pmatrix} H_+ \\ H_- \end{pmatrix} = \frac{1}{\mu_0 c}\begin{pmatrix} -\mu_0 c\tilde{T}_{xx}^\mu - in_{\mu+}^{\kappa_z}\kappa_z & 0 \\ 0 & -\mu_0 c\tilde{T}_{xx}^\mu + in_{\mu-}^{\kappa_z}\kappa_z \end{pmatrix}\begin{pmatrix} E_+ \\ E_- \end{pmatrix} \tag{70}$$

within the media $\mu = 1$ or 2, where $n_\pm$ depends on the sign $\kappa_z$, and $\kappa_z = -1$ for incident and transmitted light, while $\kappa_z = 1$ for reflected light. Here,

$$
\begin{aligned}
\tilde{T}_{xx} &= \frac{1}{3}(G_{xx} - G_{zz}) - \frac{1}{6}\omega(a'_{yzx} - a'_{zyx}) - i\left[G'_{xx} - \frac{1}{2}\omega(a_{yzx} - a_{zyx})\right] \\
&= \frac{i}{3}\sigma_{zxy} - \frac{1}{2}\sigma'_{xyz}.
\end{aligned} \tag{71}
$$

As we consider light incident from medium 1 to medium 2, the electric field in medium 1 consists of incident and reflected fields while that in medium 2 is the transmitted field.

$$
\begin{aligned}
\mathbf{E}_1 &= \mathbf{E}^i + \mathbf{E}^r \equiv (1 + r)\mathbf{E}^i, \\
\mathbf{E}_2 &= \mathbf{E}^t \equiv t\mathbf{E}^i,
\end{aligned} \tag{72}
$$

where

$$r = \begin{pmatrix} r_{++} & r_{++} \\ r_{-+} & r_{--} \end{pmatrix} = \begin{pmatrix} r_{++} & 0 \\ 0 & r_{--} \end{pmatrix}, \quad t = 1 + r \tag{73}$$

by $C_{3z}$ symmetry and the continuity of **E** at the interface.

The **H** field satisfies the boundary condition

$$\mathbf{H}^t = \mathbf{H}^i + \mathbf{H}^r + \hat{z}\times\mathbf{j}_f, \tag{74}$$

where $\hat{z}\times\mathbf{j}_f = (e^2/2\pi h)(\theta_2^{(z)} - \theta_1^{(z)})\mathbf{E}$ is the surface current due to the axion magneto-electric coupling. In terms of **B** fields, the boundary condition has the form of Eq. (13):

$$\mathbf{B}^t = \mathbf{B}^i + \mathbf{B}^r + \mu_0\hat{z}\times\mathbf{j}_s, \tag{75}$$

where $\mathbf{j}_s = \mathbf{j}_f + (\tilde{T}_{xx}^2 - \tilde{T}_{xx}^1)\mathbf{E}\times\hat{z}$ is the total two-dimensional surface current density. By solving the boundary condition, we obtain

$$
\begin{aligned}
r_{++} &= \frac{n_{1L} - n_{2L} - i\mu_0 c\sigma_{xy}^s}{n_{1R} + n_{2L} + i\mu_0 c\sigma_{xy}^s}, \\
r_{--} &= \frac{n_{1R} - n_{2R} + i\mu_0 c\sigma_{xy}^s}{n_{1L} + n_{2R} - i\mu_0 c\sigma_{xy}^s}, \\
r_{+-} &= r_{-+} = 0,
\end{aligned} \tag{76}
$$

where $n_{\mu L} = n_{\mu-+}^- = n_{\mu-}^+$ is the refractive index for left circularly polarization, and $n_{\mu R} = n_{\mu-}^- = n_{\mu+}^+$ is the refractive index for the right circularly polarization, and

$$\sigma_{xy}^s = \tilde{G}_{xx}^{(z),2} - \tilde{G}_{xx}^{(z),1} = \tilde{T}_{xx}^{\mu=2} - \tilde{T}_{xx}^{\mu=1} + \sigma_{xy}^f \tag{77}$$

is the two-dimensional surface conductivity, and $\sigma_{xy}^f = (e^2/2\pi h)(\theta_2^{(z)} - \theta_1^{(z)})$. From the expressions of $r_{++}$ and $r_{--}$ and Eq. (69), we obtain the Kerr angle

$$\varphi_K = \tan^{-1}\frac{r_{xy}}{r_{xx}} = \tan^{-1}\left[\frac{-i(r_{++} - r_{--})}{r_{++} + r_{--}}\right]. \tag{78}$$

In non-magnetic systems where $G_{ij} = a'_{ijk} = 0$, the Kerr angle vanishes systems because

$$r_{++} - r_{--} \propto i\mu_0 c(\sigma''^{\mu=2}_{xyz} - \sigma''^{\mu=1}_{xyz}) + n_{2R} - n_{2L} - (n_{1R} - n_{1L}) = 0, \quad (79)$$

where we use that $\tilde{T}^{\mu}_{xx} = -\sigma''^{\mu}_{xyz}/2$ when $\sigma_{ijk} = 0$. One may think of this cancellation as a compensation between bulk and surface responses. The refractive indices are responsible for circular birefringence in the bulk, while $\delta\sigma'_{xyz}$ is responsible for the surface current that leads to the jump of **B** field at the surface. Their effects cancel such that there is no net polar Kerr rotation (i.e., no Kerr rotation at normal incidence), compatible with the reciprocal relation imposed by time reversal symmetry[33,51–54].

In contrast, a nonzero polar Kerr rotation $PT$-symmetric antiferromagnets occurs because no compensation occurs due to the absence of circular birefringence in the bulk. By imposing $PT$ symmetry and breaking $T$ symmetry, we get a nonzero Kerr angle from

$$r_{xx} = \frac{1}{2}(r_{++} + r_{--}) = \frac{(n_1 - n_2)(n_1 + n_2) - (\mu_0 c\sigma^s_{xy})^2}{(\mu_0 c\sigma^s_{xy})^2 + (n_1 + n_2)^2},$$
$$r_{xy} = \frac{1}{2i}(r_{++} - r_{--}) = -\frac{2n_1(\mu_0 c\sigma^s_{xy})}{(\mu_0 c\sigma^s_{xy})^2 + (n_1 + n_2)^2}. \quad (80)$$

Note that our derivation using the **H** field make it manifest that the magneto-optic reflection has two different origins: bulk-propagation and surface effects, respectively contained in $T_{xx}$ and $\theta^{(z)}$.

**Reflection and transmission from two interfaces.** Now we consider three media with $(n_\mu, m_\mu)$, where $\mu = 1, 2, 3$, as shown in Fig. 3b. We consider the limit where the wavelength of light $\lambda$ is much larger than the sample thickness $d$ and neglect the variation of the electric field between the top and bottom within the sample.

The Jones reflection matrix of the sample for light incident from medium 1 is then

$$
\begin{aligned}
r &= r_T + t'_T e^{i\phi} r_B e^{i\phi} t_T + t'_T e^{i\phi} r_B (e^{i\phi} r'_T e^{i\phi} r_B) e^{i\phi} t_T + t'_T e^{i\phi} r_B (e^{i\phi} r'_T e^{i\phi} r_B)^2 e^{i\phi} t_T + \cdots \\
&= r_T + e^{2i\phi} t'_T r_B (1 - e^{2i\phi} r'_T r_B)^{-1} t_T \\
&= r_T + e^{2i\phi}(1 + r'_T) r_B (1 - e^{2i\phi} r'_T r_B)^{-1} (1 + r_T),
\end{aligned}
\quad (81)
$$

where we used $t_{T,B} = 1 + r_{T,B}$ and $t'_T = 1 + r'_T$, and

$$\phi = \frac{n_2 \omega d}{c} \quad (82)$$

is the complex-valued phase obtained by the propagation across the sample. Here, the subscript for $r$ and $t$ indicates the top and bottom of the sample, and the prime indicates the process where the light propagation is reversed (we follow the notation in ref. 13).

For transmission, the Jones matrix is

$$
\begin{aligned}
t &= t_B e^{i\phi} t_T + t_B (e^{i\phi} r'_T e^{i\phi} r_B) e^{i\phi} t_T + t_B (e^{i\phi} r'_T e^{i\phi} r_B)(e^{i\phi} r'_T e^{i\phi} r_B) t_T + \cdots \\
&= t_B (1 - e^{2i\phi} r'_T r_B)^{-1} e^{i\phi} t_T \\
&= (1 + r_B)(1 - e^{2i\phi} r'_T r_B)^{-1} e^{i\phi} (1 + r_T).
\end{aligned}
\quad (83)
$$

The expressions above show that both $r$ and $t$ can be obtained obtained from the reflective Jones matrices at the top and bottom interfaces. We suppose that each media has $PT$, $C_{3z}$-, and $M_x T$

symmetries, which is the setup in main text. Then we have

$$
\begin{aligned}
(r_T)_{xx} &= \frac{(n_1 - n_2)(n_1 + n_2) - (\mu_0 c\sigma^T_{xy})^2}{(\mu_0 c\sigma^T_{xy})^2 + (n_1 + n_2)^2}, \\
(r_T)_{xy} &= -\frac{2n_1(\mu_0 c\sigma^T_{xy})}{(\mu_0 c\sigma^T_{xy})^2 + (n_1 + n_2)^2}, \\
(r'_T)_{xy} &= \frac{-(n_1 - n_2)(n_1 + n_2) - (\mu_0 c\sigma^T_{xy})^2}{(\mu_0 c\sigma^T_{xy})^2 + (n_1 + n_2)^2}, \\
(r'_T)_{xy} &= -\frac{2n_2(\mu_0 c\sigma^T_{xy})}{(\mu_0 c\sigma^T_{xy})^2 + (n_1 + n_2)^2}, \\
(r_B)_{xx} &= \frac{(n_2 - n_3)(n_2 + n_3) - (\mu_0 c\sigma^B_{xy})^2}{(\mu_0 c\sigma^B_{xy})^2 + (n_2 + n_3)^2}, \\
(r_B)_{xy} &= \frac{2n_2(\mu_0 c\sigma^B_{xy})}{(\mu_0 c\sigma^B_{xy})^2 + (n_2 + n_3)^2},
\end{aligned}
\quad (84)
$$

where the superscript $T$ or $B$ for the surface Hall conductivity $\sigma_{xy}$ indicates the top and bottom surfaces.

**Kerr angle and Stokes parameters**

The reflective circular dichroism and Kerr rotation angle can be defined with Stokes parameters $s_{i=0,1,2,3}$ for reflected light by

$$
\begin{aligned}
\text{RCD} &\equiv \frac{s_3}{s_0}, \\
\vartheta_K &\equiv \frac{1}{2}\tan^{-1}\frac{s_2}{s_1},
\end{aligned}
\quad (85)
$$

where Stokes parameters for reflected light are

$$
\begin{aligned}
s_0 &= I^r_+ + I^r_-, \\
s_1 &= I^r_x - I^r_y, \\
s_2 &= I^r_{x+y} - I^r_{x-y}, \\
s_3 &= I^r_+ - I^r_-,
\end{aligned}
\quad (86)
$$

where $\hat{\pm} = \hat{x} \pm i\hat{y}$. While this definition of the complex Kerr angle is different from the more popular $\phi_K = r_{xy}/r_{xx} = \varphi_K + i\eta_K$ we use in the main text, it has the advantage that it can be obtained simply by measuring the intensity of the linearly polarized light. In our case where $C_{3z}$ symmetry is present, the reflective coefficients in circularly polarized basis are

$$
\begin{aligned}
r_{++} &= \frac{1}{2}\left[r_{xx} + r_{yy} + i(r_{xy} - r_{yx})\right] = r_{xx} + i r_{xy}, \\
r_{--} &= \frac{1}{2}\left[r_{xx} + r_{yy} - i(r_{xy} - r_{yx})\right] = r_{xx} - i r_{xy}, \\
r_{+-} &= \frac{1}{2}\left[r_{xx} - r_{yy} - i(r_{xy} + r_{yx})\right] = 0, \\
r_{-+} &= \frac{1}{2}\left[r_{xx} - r_{yy} + i(r_{xy} + r_{yx})\right] = 0.
\end{aligned}
\quad (87)
$$

It follows that $\text{RCD} = 2\text{Im}[r^*_{xx} r_{xy}]/(|r_{xx}|^2 + |r_{xy}|^2) \simeq 2\text{Im}[r_{xy}/r_{xx}] = 2\eta_K$ and $\vartheta_K = \frac{1}{2}\tan^{-1}[2\text{Re}(r_{xx} r^*_{yx})/(|r_{xx}|^2 - |r_{xy}|^2)] \simeq \text{Re}[r_{xy}/r_{xx}] = \varphi_K$ for small $|r_{xy}/r_{xx}|$.

**Tight-binding model of MnBi$_2$Te$_4$**

We begin with the lattice version of the three-dimensional low-energy model of MnBi$_2$Te$_4$ at $\Gamma = (0, 0, 0)$ presented in ref. 36. We consider the four basis states.

$$|P1^+_z, \uparrow\rangle, \quad |P2^-_z, \uparrow\rangle, \quad |P1^+_z, \downarrow\rangle, \quad |P2^-_z, \downarrow\rangle, \quad (88)$$

where $P1$ and $P2$ states originates from the $p$ orbitals in Bi and Te, respectively, the sign $\pm$ indicates the inversion parities, and the arrows

indicate the spin-$z$ direction. The symmetry operators in the non-magnetic state are time reversal $T$, inversion $P$, threefold rotation $C_{3z}$, and twofold rotation $C_{2x}$.

$$T = is_y K, \quad P = \tau_z, \quad C_{3z} = \exp\left[-i\frac{\pi}{3}s_z\right], \quad C_{2x} = -is_x. \tag{89}$$

Because of $PT = is_y\tau_z K$ symmetry, only the following five Gamma matrices are allowed in the Hamiltonian in addition to the overall energy shift proportional to the identity matrix $\Gamma_0$.

$$\begin{aligned}
\Gamma_1 &= s_x\tau_x(+,-),\\
\Gamma_2 &= s_y\tau_x(-,-),\\
\Gamma_3 &= s_z\tau_x(-,-),\\
\Gamma_4 &= s_0\tau_y(+,-),\\
\Gamma_5 &= s_0\tau_z(+,+).
\end{aligned} \tag{90}$$

Here, the pair signs show the commutation (+) and anticommutation (−) relations with $C_{2x}$ and $P$ in order, i.e., $(\epsilon_{C_{2x}}, \epsilon_P)$ where $M\Gamma_i = \epsilon_M\Gamma_i M$ for $i = 1, \dots, 5$.

The low-energy effective Hamiltonian up to second order in $\mathbf{k}$ has the form

$$\begin{aligned}
h_{\text{eff}} &= \epsilon(\mathbf{k})s_0\tau_0 + A_1 k_z s_z \tau_x + A_2(k_x s_x + k_y s_y)\tau_x + M(\mathbf{k})\tau_z\\
&= \epsilon(\mathbf{k})\Gamma_0 + A_1 k_z \Gamma_3 + A_2(k_x\Gamma_1 + k_y\Gamma_2) + M(\mathbf{k})\Gamma_5,
\end{aligned} \tag{91}$$

where

$$\begin{aligned}
\epsilon(\mathbf{k}) &= C_0 + C_1 k_z^2 + C_2(k_x^2 + k_y^2),\\
M(\mathbf{k}) &= M_0 + M_1 k_z^2 + M_2(k_x^2 + k_y^2).
\end{aligned} \tag{92}$$

Let us find a tight-binding Hamiltonian on the two-dimensional triangular lattice that leads to the above low-energy Hamiltonian. We suppose that each lattice site has four degrees of freedom given by Eq. (88).

$$\hat{H}_{\text{TB}} = \sum_{i,\alpha,\beta} \hat{c}_{i\alpha}^\dagger (h_0)_{\alpha\beta}\hat{c}_{i\beta} - \sum_{\langle i,j\rangle,\alpha,\beta} \hat{c}_{i\alpha}^\dagger t_{\alpha\beta}^{ij}\hat{c}_{j\beta}. \tag{93}$$

Here, the onsite Hamiltonian satisfying all the symmetries of the nonmagnetic state is

$$h_0 = e_0\Gamma_0 + e_5\Gamma_5. \tag{94}$$

Along the $z$ direction, the nearest-neighbor hopping matrices are

$$T_4 \equiv t^{j+\mathbf{a}_4, j} = t_0^z\Gamma_0 + it_3^z\Gamma_3 + t_5^z\Gamma_5, \tag{95}$$

where $\mathbf{a}_4 = (0, 0, a_z)$, $a_z$ is the inter-layer lattice parameter. the form of $T_4$ is constrained by the following symmetry conditions

$$\begin{aligned}
T T_4 T^{-1} &= T_4,\\
C_{3z} T_4 C_{3z}^{-1} &= T_4,\\
C_{2x} T_4 C_{2x}^{-1} &= T_4^\dagger,\\
P T_4 P^{-1} &= T_4^\dagger,
\end{aligned} \tag{96}$$

For the in-plane directions, the hopping matrices are

$$\begin{aligned}
T_1 &\equiv t^{j+\mathbf{a}_1, j} = t_0\Gamma_0 + it_1\Gamma_1 + it_4\Gamma_4 + t_5\Gamma_5 = (t^{j,j+\mathbf{a}_1})^\dagger,\\
T_2 &\equiv t^{j+\mathbf{a}_2, j} = C_{3z}T_1 C_{3z}^{-1} = t_0\Gamma_0 + it_1\frac{-\Gamma_1 + \sqrt{3}\Gamma_2}{2} + it_4\Gamma_4 + t_5\Gamma_5 = (t^{j,j+\mathbf{a}_2})^\dagger,\\
T_3 &\equiv t^{j+\mathbf{a}_3, j} = C_{3z}T_2 C_{3z}^{-1} = t_0\Gamma_0 + it_1\frac{-\Gamma_1 - \sqrt{3}\Gamma_2}{2} + it_4\Gamma_4 + t_5\Gamma_5 = (t^{j,j+\mathbf{a}_3})^\dagger
\end{aligned} \tag{97}$$

along the in-plane directions, where $\mathbf{a}_1 = (a, 0, 0)$, $\mathbf{a}_2 = C_{3z}\mathbf{a}_1$, $\mathbf{a}_3 = C_{3z}\mathbf{a}_2$, $a$ is the in-plane lattice parameter, and the form of $T_1$ is constrained by the following symmetry conditions

$$\begin{aligned}
T T_1 T^{-1} &= T_1,\\
C_{2x} T_1 C_{2x}^{-1} &= T_1,\\
P T_1 P^{-1} &= T_1^\dagger,
\end{aligned} \tag{98}$$

and $T_i$s are related by $C_{3z}$ because of $C_{3z}$ symmetry imposing, for example,

$$\begin{aligned}
\hat{c}_{j+\mathbf{a}_2,\alpha}^\dagger (T_2)_{\alpha\beta}\hat{c}_{j\beta} &= \hat{C}_{3z}\hat{c}_{j+\mathbf{a}_1,\alpha}^\dagger (T_1)_{\alpha\beta}\hat{c}_{j\beta}\hat{C}_{3z}^{-1}\\
&= \left(\hat{C}_{3z}\hat{c}_{j+\mathbf{a}_1,\alpha}^\dagger\hat{C}_{3z}^{-1}\right)(T_1)_{\alpha\beta}\left(\hat{C}_{3z}\hat{c}_{j\beta}\hat{C}_{3z}^{-1}\right)\\
&= \hat{c}_{j+\mathbf{a}_2,\gamma}^\dagger (C_{3z})_{\gamma\alpha}(T_1)_{\alpha\beta}\hat{c}_{j\delta}(C_{3z}^*)_{\delta\beta}\\
&= \hat{c}_{j+\mathbf{a}_2,\alpha}^\dagger (C_{3z}T_1 C_{3z}^{-1})_{\alpha\beta}\hat{c}_{j\beta}.
\end{aligned} \tag{99}$$

In momentum space, the tight-binding Hamiltonian becomes

$$\begin{aligned}
h_{\text{TB}} &= h_0 - (T_1 e^{-ik_1 a} + T_2 e^{-ik_2 a} + T_3 e^{-ik_3 a} + T_4 e^{-ik_4 a_z} + h.c.)\\
&= \left[e_0 - 2t_0(\cos k_1 a + \cos k_2 a + \cos k_3 a) - 2t_0^z\cos k_4 a_z\right]\Gamma_0\\
&\quad - t_1(2\sin k_1 a - \sin k_2 a - \sin k_3 a)\Gamma_1 - \sqrt{3}t_1(\sin k_2 a\\
&\quad - \sin k_3 a)\Gamma_2 - 2t_3^z\sin k_4 a_z\Gamma_3\\
&\quad - 2t_4(\sin k_1 a + \sin k_2 a + \sin k_3 a)\Gamma_4\\
&\quad + \left[e_5 - 2t_5(\cos k_1 a + \cos k_2 a + \cos k_3 a) - 2t_5^z\cos k_4 a_z\right]\Gamma_5,
\end{aligned} \tag{100}$$

where

$$\begin{aligned}
k_1 &= k_x,\\
k_2 &= \frac{1}{2}\left(-k_x + \sqrt{3}k_y\right),\\
k_3 &= \frac{1}{2}\left(-k_x - \sqrt{3}k_y\right),\\
k_4 &= k_z.
\end{aligned} \tag{101}$$

By expanding the tight-binding Hamiltonian up to second order in $\mathbf{k}$, we obtain

$$\begin{aligned}
h_{\text{TB}} &= \left[e_0 - 6t_0 - 2t_0^z + \frac{3}{2}t_0 a^2(k_x^2 + k_y^2) + t_0^z a_z^2 k_z^2\right]\Gamma_0\\
&\quad - 3t_1 a k_x\Gamma_1 - 3t_1 a k_y\Gamma_2 - 2t_3^z a k_z\Gamma_2\\
&\quad + \left[e_5 - 6t_5 - 2t_5^z + \frac{3}{2}t_5 a^2(k_x^2 + k_y^2) + t_5^z a_z^2 k_z^2\right]\Gamma_5 + O(k^3).
\end{aligned} \tag{102}$$

Comparing this with $h_{\text{eff}}$, we find

$$\begin{aligned}
e_0 &= C_0 + 2C_1/a_z^2 + 4C_2/a^2,\\
e_5 &= M_0 + 2M_1/a_z^2 + 4M_2/a^2,\\
t_0 &= \frac{2C_2}{3a^2},\\
t_0^z &= \frac{C_1}{a_z^2},\\
t_1 &= -\frac{A_2}{3a},\\
t_3^z &= -\frac{A_1}{2a_z},\\
t_5 &= \frac{2M_2}{3a^2},\\
t_5^z &= \frac{M_1}{a_z^2}.
\end{aligned} \tag{103}$$

We use the parameters derived in ref. 36 with a modification, $C_2 = 0$, we take in order to obtain an insulating filling:

$$
\begin{aligned}
C_0 &= -0.0048 \text{ eV}, \\
C_1 &= 2.7232 \text{ eVÅ}^2, \\
C_2 &= 0 \text{ eVÅ}^2, \\
M_0 &= -0.1165 \text{ eV}, \\
M_1 &= 11.9048 \text{ eVÅ}^2, \\
M_2 &= 9.4048 \text{ eVÅ}^2, \\
A_1 &= 2.7023 \text{ eVÅ}, \\
A_2 &= 3.1964 \text{ eVÅ}, \\
a &= 4.334 \text{ Å}, \\
a_z &= \tfrac{1}{3}c = \tfrac{1}{3}40.91 \text{ Å} = 13.64 \text{ Å}.
\end{aligned}
\tag{104}
$$

The spin part is described by

$$
h_{\text{spin}} = -\mu_B \mathbf{B} \cdot \mathbf{s}\tau_0,
\tag{105}
$$

where the Bohr magneton is

$$
\mu_B = 3.8099 \frac{e}{\hbar} \text{ eVÅ}^2.
\tag{106}
$$

We consider the antiferromagnetic state of a few-layer $MnBi_2Te_4$ with layer-alternating out-of-plane moments. The antiferromagnetic moment is described by

$$
h_{\text{AFM}} = m l_z s_z \tau_0,
\tag{107}
$$

where $l_z$ is a Pauli matrix in the sublattice (i.e., layer) space. We use $m = 0.03$ eV for our calculations. The full $8 \times 8$ tight-binding Hamiltonian in momentum space is then

$$
\begin{aligned}
h_{\text{TB}} &= h_{\text{para}} + h_{\text{AFM}} \\
&= [e_0 - 2t_0(\cos k_1 a + \cos k_2 a + \cos k_3 a)]l_0\Gamma_0 - 2t_0^z \cos k_4 a_z l_x \Gamma_0 \\
&\quad - t_1(2\sin k_1 a - \sin k_2 a - \sin k_3 a)l_0\Gamma_1 - \sqrt{3}t_1(\sin k_2 a \\
&\quad - \sin k_3 a)l_0\Gamma_2 + 2t_3^z \sin k_4 a_z l_y \Gamma_3 - 2t_4(\sin k_1 a + \sin k_2 a + \sin k_3 a)l_0\Gamma_4 \\
&\quad + [e_5 - 2t_5(\cos k_1 a + \cos k_2 a + \cos k_3 a)]\Gamma_5 - 2t_5^z \cos k_4 a_z l_x \Gamma_5 + m l_z \Gamma_{12},
\end{aligned}
\tag{108}
$$

where $\Gamma_{12} = \Gamma_1 \Gamma_2 / 2i = s_z \tau_0$.

## Data availability
The data that support the findings of this study are available in the main text and Supplementary Information. Further information is available from the corresponding author upon reasonable request.

## Code availability
The codes that support the findings of this study are available from the corresponding author upon reasonable request.

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

## Acknowledgements
We appreciate Jian-Xiang Qiu, Philip Kim, Ari Turner, Ivo Souza, and Allan MacDonald for helpful discussions. This work was supported by the Center for Advancement of Topological Semimetals, an Energy Frontier Research Center funded by the U.S. Department of Energy Office of Science, Office of Basic Energy Sciences, through the Ames Laboratory under contract No. DE-AC02-07CH11358 (J.A., S.-Y.X., and A.V.).

## Author contributions
J.A. performed theoretical and numerical analysis. J.A., S.-Y.X., and A.V. contributed to the development of original ideas, discussed the results, and wrote the manuscript.

## Competing interests
The authors declare no competing interests.
