## [Peer Review File · Nature Communications]

REVIEWER COMMENTS

Reviewer #1 (Remarks to the Author):

The manuscript presents a theoretical derivation of the optical axion electrodynamics and Kerr effect. By considering the electric quadrupole/magnetic dipole order interactions with light, the work provides general formulas of magneto-electric couplings and axion optical angles. Finally, the authors use the even-layer MnBi_2Te_4 (MBT), as an example to demonstrate their theoretical models. Particularly, they show that the Kerr effect is significant while the Faraday effect is weak in axion insulators.

The derivation seems fine. However, I must admit that it is hard for me to follow all details. On the other hand, I have a few questions and comments about the applications of the formulas to the MBT material.

1. I am not sure the validity of the tight-binding model in studying MBT. First, this is a low-energy model. According to my impression, it is valid for the band structures near the band gap (~ 0.2 eV). Thus, the discussion within the low energy range in Figures 4 and 5 is fine while is questionable for the result around 1-2 eV. Moreover, I am not sure if the Hamiltonian in Eq. 18 is for bulk or two-dimensional MBT structures. In other words, does Eq. 18 capture the change of the band structure according to the thickness? Eq. 11 needs the transition energy. Thus, the variation of band structure and band gap may strongly impact the final results in Figures 4 and 5. It may be necessary to add the band structures of few-layers and bulk structures in the supplementary information.

2. When discussing Figure 5, the authors claim that the calculated Kerr rotation angle can reach 0.2 degree at photon energy larger than 1 eV. Unfortunately, I cannot read this value from Figure 5 (a). It seems that the value is about 0.02 degree. Moreover, as I mentioned in the above question, the low-energy Hamiltonian may not give reliable answers for such high energy photons.

3. Finally, I have a comment about the length of the manuscript. It seems much longer than the length limit of nature Comm. ($\sim 5,000$ words) The authors can consider moving a substantial part of the math derivation into the supplementary information. This will also help the general readers to focus on the main physics picture and results.

Reviewer #3 (Remarks to the Author):

In this manuscript, the authors study the optical response of layered three-dimensional (3D) systems, focusing on the surface contribution to the optical Kerr and Faraday effects. Starting from the fact that an electric current is related to macroscopic quantities such as electric dipole, electric quadrupole, and magnetic dipole (corresponding to a polarization current and a magnetization current), the authors derive a surface current density of a layered 3D system. From this surface current, the authors define a linear (and diagonal) magnetoelectric coupling coefficient by $\partial P_i / \partial B_i$, which is called the optical axion angle in this manuscript [Eqs. (4) and (5)]. Then the authors apply the obtained formulae to antiferromagnetic topological insulators such as MnBi₂Te₄ with a finite thickness.

3D antiferromagnetic topological insulators such as MnBi₂Te₄ have recently been experimentally realized and have attracted much attention in the topological materials community, as a promising platform for the observation of the topological (quantized) magnetoelectric effect, i.e., the axion electrodynamics. The present manuscript reveals another (experimentally accessible) contribution to the optical response of MnBi₂Te₄. Given these aspects, the present manuscript can be of interest to the community.

However, I am afraid that the manuscript has a limited scope and exaggerates the obtained results. My understanding is that the action of the form $\mathbf{E} \cdot \mathbf{B}$ (i.e., the theta term) is exact in the field theory literature. In this regard, I am not convinced by the authors' terminology "optical axion electrodynamics". In my view, the authors investigate a magnetoelectric (magneto-optic) phenomenon that is linear in the electric field E and in the magnetic field B , rather than the phenomenon originating from a topological field-theoretic action. In other words, linear magnetoelectric phenomena can also occur in non-topological systems. How can the authors justify that the phenomenon the authors found has a topological origin? In fact, the authors have not derived microscopically (even from a model system) the effective action of the form of the theta term which describes the phenomenon the authors found.

To summarize, in my opinion, the manuscript is suitable for publication in a more specialized journal.

This is a review of manuscript by Ahn et al., titled “Theory of optical axion electrodynamics and application to the Kerr effect in topological antiferromagnets”. The topic of the study is interesting from the theoretical point of view, and may even generate some experimental activity. I would say that the manuscript lost some clarity because of the language the authors chose to present their findings. While I certainly do not mean to tell the authors how to write their papers, for the purpose of this review, and my own understanding, I would like to rephrase the discussion a bit. I would like to make several statements, and the authors could comment on whether they are true or not.

1. The authors consider the effects of spatial dispersion of conductivity, which are linear in the gradients of the electric field.
2. The separation of the medium response in electric and magnetic parts is ambiguous at finite ω and \mathbf{k} , because the Faraday’s law allows to express the magnetic field in terms of the electric field at finite ω and \mathbf{k} .
3. Instead of introducing several material tensors the way authors do, one can simply describe the current response of the medium with a single conductivity tensor, $J_i = \sigma_{ij}(\omega, \mathbf{k})E_j$.
4. The conductivity tensor can be expanded to linear order in \mathbf{k} : $\sigma_{ij} = \sigma_{ij}(\omega) + \tilde{\sigma}_{ijl}(\omega)k_l$, and the entire discussion of the paper is just the one of $\tilde{\sigma}_{ijk}(\omega)$.
5. $\tilde{\sigma}_{ijk}(\omega)$ has well known physical content: its part symmetric in the first two indices describes the gyrotropic birefringence, as described by Hornreich and Shtrikman, in “Theory of Gyrotropic Birefringence”, Phys. Rev. 171, 1065 (1968). The antisymmetric part describes the natural optical activity.

If the above statements are correct, the authors should substantiate the need for the introduction of new names for the effects being considered. How are they different from the well-known ones, mentioned above? It seems to me that the authors just call the absorptive and reactive parts of the gyrotropic birefringence and natural optical activity with some new names, but there is hardly need for that. There is also modern band theory of both, developed in Refs 19, 48, 50, 51 of the paper. If there are mistakes or omissions in those works, the authors should discuss the specific places where those are in previous works.

Furthermore, it would be very helpful to discuss the electromagnetic boundary conditions the authors used. It is well known that in the presence of spatial dispersion, one has to add effective surface conductivities near sample boundaries, which crucially affect the optical properties of samples. This was shown by Agranovich and Yudson in “On phenomenological electrodynamics of gyrotropic media”, Opt. Commun. 9, 58 (1973). I strongly suspect that using the proper boundary conditions will actually cancel the Kerr effect that the authors claim to come from the spatial dispersion of conductivity. The modified boundary conditions might have been alluded to in the sentence “While electric quadrupole and magnetic dipole moments do not generate macroscopic currents in macroscopically homogeneous lattice systems, they generate currents on the system boundary where the material property”. But it would be useful to see the boundary conditions explicitly in terms of tensors \mathbf{a} and \mathbf{G} .

I am looking forward to the authors’ response to these questions. As a final minor comment, I would like to mention that the name “optical axion electrodynamics” does not seem to make sense, electrodynamics is just that, it cannot be optical, or infrared. Perhaps “axion electrodynamics at finite frequencies” is fine, but as explained above, this seems to be repackaging of known physical phenomena.

Response to Reviewer #1

Comment: The manuscript presents a theoretical derivation of the optical axion electrodynamics and Kerr effect. By considering the electric quadruple/magnetic dipole order interactions with light, the work provides general formulas of magneto-electric couplings and axion optical angles. Finally, the authors use the even-layer MnBi₂Te₄ (MBT), as an example to demonstrate their theoretical models. Particularly, they show that the Kerr effect is significant while the Faraday effect is weak in axion insulators.

The derivation seems fine. However, I must admit that it is hard for me to follow all details. On the other hand, I have a few questions and comments about the applications of the formulas to the MBT material.

Response: We thank the reviewer for carefully reading our manuscript. We have clarified various points in the revision, so we believe that our revised manuscript is more readable. We address all the questions and comments below.

Comment: 1. I am not sure the validity of the tight-binding model in studying MBT. First, this is a low-energy model. According to my impression, it is valid for the band structures near the band gap (~0.2 eV). Thus, the discussion within the low energy range in Figures 4 and 5 is fine while is questionable for the result around 1-2 eV.

Response: We thank the reviewer for this clarifying question. We agree with the reviewer that the low-energy tight-binding model loses its validity at photon energies far above the band gap. The purpose of our model calculations is proof of principle (such as significant Kerr and negligible Faraday effect) and retains qualitative features, such as symmetries, but is not meant to capture quantitative features of MnBi₂Te₄. A full-scale DFT calculation, combined with experimental discoveries, is on its way with collaborators, but the results are qualitatively similar. To avoid confusion, we have revised the manuscript such that it includes

“The goal of our calculations here is to cement the validity of our new theory by providing a concrete model example as well as to understand qualitative features (e.g., the dominance of the axion contribution and the significant Kerr and negligible Faraday effects) of the magneto-optic response in MnBi₂Te₄. Our model is expected to quantitatively capture the low-energy properties of the material. On the other hand, at photon energies much larger than the band gap, a precise quantitative calculation of the magneto-optical spectrum will require a model, like the full ab-initio model, that captures all the significant optical transitions involving those states neglected in our model.”.

Comment: Moreover, I am not sure if the Hamiltonian in Eq. 18 is for bulk or two-dimensional MBT structures. In other words, does Eq. 18 capture the change of the band structure according to the thickness? Eq. 11 needs the transition energy. Thus, the variation of band structure and band gap may strongly impact the final results in Figures 4 and 5. It may be necessary to add the band structures of few-layers and bulk structures in the supplementary information.

Response: The layer-number variation of the band structure and band gap indeed affects the response properties, especially at low energies around and below the gap. For example, figure 1 above shows that the axion angle has a peak at the surface gap, around 0.05~0.1 eV, whose value depends on the number of layers (cf. main-Fig. 4 and SI-Fig. 2). This variation is captured by the Hamiltonian in Equation (18) because it is a general form that can describe the bulk or two-dimensional MBT depending on how we impose the boundary condition for the lattice. We have added the layer-dependent band structure in Supplementary Information by following the suggestion of the reviewer.

Figure 1. Axion coupling vs band structure.

Comment: 2. When discussing Figure 5, the authors claim that the calculated Kerr rotation angle can reach 0.2 degree at photon energy larger than 1 eV. Unfortunately, I cannot read this value from Figure 5 (a). It seems that the value is about 0.02 degree. Moreover, as I mentioned in the above question, the low-energy Hamiltonian may not give reliable answers for such high energy photons.

Response: We appreciate the reviewer for the careful proofreading. Actually, the 0.2 degree was a typo. The Kerr angle is about 0.02 degree in our low-energy model. On the other hand, Kerr angle in more realistic DFT calculations that give a much larger value of about 0.2 degree – which we do not show in the current paper.

We have revised the relevant sentence to “*The calculated Kerr rotation angle φ_K is about 0.02 degree at photon energies larger than 1 eV, which is about one order of magnitude smaller than $\varphi_K < \sim 1$ degree in typical ferromagnets although our antiferromagnetic system has zero net magnetic moment. The Kerr angle in real MnBi2Te4 can even be much enhanced because of the contributions from higher-energy bands that we do not include here.*”

Comment: 3. Finally, I have a comment about the length of the manuscript. It seems much longer than the length limit of nature Comm. (~5,000 words) The authors can consider moving a substantial part of the math derivation into the supplementary information. This will also help the general readers to focus on the main physics picture and results.

Response: We agree that it is good to focus on the main physical pictures. However, we have already put most of the derivations in Methods and kept only essential parts in the main text, so we do not agree that our manuscript is too long. Our estimate of the word count is about 5,000 words for the main text, which is within the length limit of Nature Communications. We believe that our manuscript format agrees with the journal’s style that allows the presentation of an in-depth study. We will consult with the editors about whether the length needs revision.

Response to Reviewer #2

Comment: This is a review of manuscript by Ahn et al., titled “Theory of optical axion electrodynamics and application to the Kerr effect in topological antiferromagnets”. The topic of the study is interesting from the theoretical point of view, and may even generate some experimental activity.

Response: We thank the referee for acknowledging that our work is interesting from both experimental and theoretical points of view.

Comment: I would say that the manuscript lost some clarity because of the language the authors chose to present their findings. While I certainly do not mean to tell the authors how to write their papers, for the purpose of this review, and my own understanding, I would like to rephrase the discussion a bit. I would like to make several statements, and the authors could comment on whether they are true or not.

1. The authors consider the effects of spatial dispersion of conductivity, which are linear in the gradients of the electric field.
2. The separation of the medium response in electric and magnetic parts is ambiguous at finite ω and k , because the Faraday’s law allows to express the magnetic field in terms of the electric field at finite ω and k .
3. Instead of introducing several material tensors the way authors do, one can simply describe the current response of the medium with a single conductivity tensor, $J_i = \sigma_{ij}(\omega, k)E_j$.
4. The conductivity tensor can be expanded to linear order in k : $\sigma_{ij} = \sigma_{ij}(\omega) + \tilde{\sigma}_{ijl}(\omega)k_l$, and the entire discussion of the paper is just the one of $\tilde{\sigma}_{ijk}(\omega)$.
5. $\tilde{\sigma}_{ijk}(\omega)$ has well known physical content: its part symmetric in the first two indices describes the gyrotropic birefringence, as described by Hornreich and Shtrikman, in “Theory of Gyrotropic Birefringence”, Phys. Rev. 171, 1065 (1968). The antisymmetric part describes the natural optical activity.

If the above statements are correct, the authors should substantiate the need for the introduction of new names for the effects being considered. How are they different from the well-known ones, mentioned above? It seems to me that the authors just call the absorptive and reactive parts of the gyrotropic birefringence and natural optical activity with some new names, but there is hardly need for that. There is also modern band theory of both, developed in Refs 19, 48, 50, 51 of the paper. If there are mistakes or omissions in those works, the authors should discuss the specific places where those are in previous works.

Response: We thank the reviewer for helpful comments that point out a possible point of confusion. We would like to emphasize that axion electrodynamics is distinct from gyrotropic birefringence and natural optical activity, and its description needs a different theoretical approach that we take in the current work.

We agree with all statements from 1 to 5 if it is regarding the bulk response. However, the very interesting part of axion electrodynamics is that it is not accounted for by the current response in the bulk. This means that the axion part is not seen by assuming fully periodic

boundary conditions and taking Fourier transformation to the 3D momentum space, as in done in statement 3 of the reviewer. We can observe this in the expression $\tilde{\sigma}_{ijk}(\omega) = ic(\epsilon_{jlm}G_{im} + \epsilon_{ilm}G_{jm}) + \omega a'_{ijk}$ that relates the conductivity tensor with magneto-electric coupling G_{ij} and the electric quadrupole susceptibility a'_{ijk} . One can check that the trace of G has no contribution to $\tilde{\sigma}_{ijk}(\omega)$ by inserting $G_{im} = \delta_{im}\text{Tr}G/3 + (\text{traceless parts})$ in the equation above. As the axion part (i.e., the trace part) of the magneto-electric coupling is missing in $\tilde{\sigma}_{ijk}(\omega)$, the axion magneto-electric effect should be studied independently. This property contrasts with gyrotropic birefringence and natural optical activity, which comes from the propagation of light modified within the bulk and thus captured by $\tilde{\sigma}_{ijk}(\omega)$ – Refs. 19, 48, 50, and 51 study these effects.

In fact, in the outlook paragraph of Ref. 19 [Malashevich & Souza PRB 82, 245118 (2010)], the authors themselves also highlighted the very fact that axion magneto-electric effects do not appear in the approach based on the bulk response. Let us directly quote the following sentences from Ref. 19:

“... The reason why the latter (←meaning axion part) is not recovered from the present formalism is that our starting point is the current response of an infinite medium to an electromagnetic wave while the trace of the ME tensor, known as the axion contribution, only affects electrodynamics at boundaries. The calculation of the axion piece at finite frequencies remains an open problem.”

The last sentence of the quote clearly indicates a need for developing a new theory for optical axion electrodynamics. It is precisely the contribution of our paper to introduce such a theory.

We have clarified the comparison to previous works by adding a new sentence “... previous studies focusing on bulk magneto-electric effect did not capture optical axion electrodynamics. ...” below Eq. (5) of the revised main text and included a review on the bulk current response in Supplementary Note 1.

Comment: Furthermore, it would be very helpful to discuss the electromagnetic boundary conditions the authors used. It is well known that in the presence of spatial dispersion, one has to add effective surface conductivities near sample boundaries, which crucially affect the optical properties of samples. This was shown by Agranovich and Yudson in “On phenomenological electrodynamics of gyrotropic media”, Opt. Commun. 9, 58 (1973). I strongly suspect that using the proper boundary conditions will actually cancel the Kerr effect that the authors claim to come from the spatial dispersion of conductivity. The modified boundary conditions might have been alluded to in the sentence “While electric quadrupole and magnetic dipole moments do not generate macroscopic currents in macroscopically homogeneous lattice systems, they generate currents on the system boundary where the material property”. But it would be useful to see the boundary conditions explicitly in terms of tensors a and G .

Response: As we describe above, what distinguishes our work from previous works is an accurate account of the surface conductivity induced by the magneto-electric effect. We imposed precise boundary conditions considering this surface conductivity when we derive the Kerr rotation angle [see the paragraph below Eq. (12)]. The boundary conditions we have to care about are those of electromagnetic fields given by Maxwell’s equations, rather than the response functions – for these, we only need that they change abruptly at the interface of two media. More explicitly, the electromagnetic boundary conditions we use are those of in-plane fields – $\mathbf{E}_{\parallel}(1) = \mathbf{E}_{\parallel}(2)$ and

$\mathbf{B}_{\parallel}(1) = \mathbf{B}_{\parallel}(2) + \mu_0 \hat{n} \times \mathbf{j}^s$ at the interface of medium 1 and medium 2 (\mathbf{j}^s is the two-dimensional surface current density), which are respectively obtained by taking the limit of infinitesimal interface thickness of the Faraday's law $\nabla \times \mathbf{E} = -\partial_t \mathbf{B}$ and Ampere's law $\nabla \times \mathbf{B} = \mu_0 (\mathbf{J} + \epsilon_0 \partial_t \mathbf{E})$. By solving these boundary condition equations, we obtain a nonzero polar Kerr angle in antiferromagnets. As a sanity check, the same formalism gives zero polar Kerr angle in a non-magnetic gyrotropic medium, which is consistent with the constraint that the polar Kerr angle vanishes by time-reversal symmetry [as derived in Agranovich and Yudson, Opt. Commun. 5, 422 (1972) and Opt. Commun. 9, 58(1973).].

While the reviewer suspects that using a proper boundary condition will cancel the Kerr effect, it is not the case in magnetic systems. A simple counterexample is given by the low-energy Kerr effect in a topological axion insulator, in which there is no bulk state that cancels the response of topological surface states. At finite optical frequencies also, there is no general principle ensuring that the surface and bulk responses cancel in magneto-electric media – and they do not cancel indeed. It is generally accepted that a nonzero Kerr effect can occur due to magneto-electric effects, while the axion contribution has not been investigated in detail.

This situation contrasts with the case of a non-magnetic gyrotropic medium, where the surface and bulk contributions have to cancel such that no Kerr effect occurs at normal incidence because of time reversal symmetry. The cancellation is directly observed by calculating the Kerr angle with the boundary conditions we impose. The circular birefringence in the bulk cancels the boundary Hall response in non-magnetic gyrotropic systems (cf. the bulk circular birefringence is forbidden in PT-symmetric antiferromagnets, so there is no way to cancel the boundary Hall response from the bulk). This is what is shown by Agranovich and Yudson in Opt. Commun. 5, 422 (1972) and Opt. Commun. 9, 58(1973).

In the revised main text, we have clarified our boundary condition by expressing it in terms of B fields instead of H fields, which greatly simplifies the discussion of the boundary condition. We have also added explicit calculations of the zero Kerr effect in a non-magnetic gyrotropic medium in Methods [see Eqs. (65-77)].

Comment: I am looking forward to the authors' response to these questions. As a final minor comment, I would like to mention that the name “optical axion electrodynamics” does not seem to make sense, electrodynamics is just that, it cannot be optical, or infrared. Perhaps “axion electrodynamics at finite frequencies” is fine, but as explained above, this seems to be repackaging of known physical phenomena.

Response: The reviewer made a fair point. However, we want to keep the term “optical axion electrodynamics” because axion electrodynamics is typically used for topological magneto-electric response, which is a static-limit response. As we explained above, there was a knowledge gap for theoretically understanding the axion part of the magneto-electric response at optical frequencies beyond the band gap, which we now fill in our work. Therefore, the term “optical axion electrodynamics” suits our finding, in our opinion.

Response to Reviewer #3

Comment: In this manuscript, the authors study the optical response of layered three-dimensional (3D) systems, focusing on the surface contribution to the optical Kerr and Faraday effects. Starting from the fact that an electric current is related to macroscopic quantities such as electric dipole, electric quadrupole, and magnetic dipole (corresponding to a polarization current and a magnetization current), the authors derive a surface current density of a layered 3D system. From this surface current, the authors define a linear (and diagonal) magnetoelectric coupling coefficient by $\partial P_i / \partial B_j$, which is called the optical axion angle in this manuscript [Eqs. (4) and (5)]. Then the authors apply the obtained formulae to antiferromagnetic topological insulators such as MnBi₂Te₄ with a finite thickness.

3D antiferromagnetic topological insulators such as MnBi₂Te₄ have recently been experimentally realized and have attracted much attention in the topological materials community, as a promising platform for the observation of the topological (quantized) magnetoelectric effect, i.e., the axion electrodynamics. The present manuscript reveals another (experimentally accessible) contribution to the optical response of MnBi₂Te₄. Given these aspects, the present manuscript can be of interest to the community.

Response: We thank the referee for insightful comments acknowledging that our study can be of interest to the topological materials community. In fact, however, the impact of our study is not limited to topological materials or, more specifically, axion insulators, as we detail below.

Comment: However, I am afraid that the manuscript has a limited scope and exaggerates the obtained results. My understanding is that the action of the form $\mathbf{E} \cdot \mathbf{B}$ (i.e., the theta term) is exact in the field theory literature. In this regard, I am not convinced by the authors' terminology "optical axion electrodynamics". In my view, the authors investigate a magnetoelectric (magneto-optic) phenomenon that is linear in the electric field E and in the magnetic field B , rather than the phenomenon originating from a topological field-theoretic action. In other words, linear magnetoelectric phenomena can also occur in non-topological systems. How can the authors justify that the phenomenon the authors found has a topological origin? In fact, the authors have not derived microscopically (even from a model system) the effective action of the form of the theta term which describes the phenomenon the authors found.

Response: We thank the reviewer for the comments that help us improve our manuscript. The reviewer raises two important concerns: (1) we exaggerate our results with inappropriate use of the term "optical axion electrodynamics", and (2) the results have a limited scope of interest. We realize that both concerns arose because our presentation failed to explain and highlight the breakthroughs clearly. Below we address the concerns by clarifying the novelty and potential impact of our work.

- (1) Our introduction of the term "optical axion electrodynamics" is justified by a) the important theoretical progress we make and b) the generally non-quantized nature of axion electrodynamics.

- a) We solve an important open question on the optical axion magneto-electric effect. The optical magnetoelectric effect is a topic extensively studied in theory. Despite the extensive studies, calculating the optical axion magneto-electric coupling remained an open question. The other components of the optical axion magneto-electric coupling are understood by considering the bulk current response as shown by Malashevich and Souza in *Phys. Rev. B* **82**, 245118 (2010). However, the axion contribution is not described it. Let us quote from their paper:

“... The reason why the latter (← meaning axion part) is not recovered from the present formalism is that our starting point is the current response of an infinite medium to an electromagnetic wave while the trace of the ME tensor, known as the axion contribution, only affects electrodynamics at boundaries. The calculation of the axion piece at finite frequencies remains an open problem.”

We have resolved this issue by developing a theory of optical axion electrodynamics.

- b) The reviewer insightfully pointed out that the term axion electrodynamics is often used to refer to topological magneto-electric response, which we completely agree with. However, more precisely, axion electrodynamics should be viewed more broadly as the electrodynamics modified by the axion magneto-electric effect (i.e., the effect of the $\theta \cdot \mathbf{E} \cdot \mathbf{B}$ term in the Lagrangian. We note that the field-theoretical description based on the $\mathbf{E} \cdot \mathbf{B}$ term is equivalent to the linear response theory description of the isotropic magneto-electric effect [e.g., Malashevich et al., *New J. Phys.* **12** 053032 (2010)]) which can be obtained by deriving the Euler-Lagrange equations from the field theoretical Lagrangian. The quantization of the θ angle is not a prerequisite for axion electrodynamics.
- (2) Our theoretical discovery is broadly relevant to a wide class of PT-symmetric antiferromagnets, providing a previously unavailable probe of fully compensated antiferromagnetic order that is highly desirable in antiferromagnetic spintronics. The non-quantized nature of optical axion electrodynamics does not indicate that our work has a limited scope. Rather, it means that our discovery can be applied to a wide class of PT-symmetric antiferromagnets that may be topologically trivial. In particular, antiferromagnets have recently attracted great interest for their prospect of robust magnetic storage because their zero net magnetization means that antiferromagnetic domains are robust against perturbing magnetic fields. However, the lack of net magnetization also means that it is hard to detect and manipulate antiferromagnetism. Our theoretical formulation of optical axion electrodynamics leads to a concrete understanding of the Kerr effect in fully compensated antiferromagnets (in the absence of magnetization) and thus sets the theoretical ground for the optical detection of antiferromagnetism. This work is relevant not only to MnBi₂Te₄, but also to other PT-symmetric antiferromagnets such as Cr₂O₃, CrI₃, etc. We may even think about optically manipulating the antiferromagnetic order using this effect. In fact, we have exciting experimental results related to this theory paper also under review. The proposal for novel optical detection (and even manipulation) of antiferromagnetic order enabled by our theoretical work clearly justifies the broad significance of our work, making it critically relevant to antiferromagnetic spintronics and

quantum magnets. From the theoretical point of view the calculation of the axion electrodynamics at finite frequencies presents a very interesting ab-initio problem. The absence of quantization implies that one must compute a number that can be compared with experiment as a function of light frequency. At the same time, the subtle nature of the axion term, wherein its effects are manifested at boundaries alone, makes such calculations challenging even in principle. We have sorted out the nontrivial conceptual aspects in our present work opening the door to future studies.

To reflect the novelty and significance that we outlined above, we have significantly revised our introduction of the paper. We thank the reviewer again for the comments that inspired us to improve our paper.

Comment: To summarize, in my opinion, the manuscript is suitable for publication in a more specialized journal.

Response: We hope that our detailed response above and improvements to the manuscript will convince the reviewer that our paper deserves publication in a prestigious journal with a broad readership, Nature Communications.

Summary of Main Changes in the Revised Manuscript

1. We have added sentences in the introduction that 1. clarify the difference of optical axion electrodynamics and the existing theory of gyrotropic birefringence/natural optical activity and 2. highlight the broad potential impact of our work.
2. We have added a sentence below Eq. (5) stating that previous works on gyrotropic birefringence did not capture optical axion electrodynamics.
3. We have expanded the discussion on Eq. (16) to emphasize its nontrivial property.
4. We have corrected the typo in the Kerr angle (0.2 degree \rightarrow 0.02 degree).
5. We have added a sentence stating that the Kerr angle calculated with our low-energy model can underestimate the Kerr angle in real MnBi₂Te₄.
6. We have simplified the discussion of the boundary condition for the magnetic field below Eq. (12). Figure 3 has been changed accordingly.
7. We have modified a model parameter to get an insulating filling at the Fermi level. This leads to slight modifications Figures 4 and 5, which do not affect key results.
8. We have added a reference in the main text:
 - a. [27] J. Orenstein, Phys. Rev. Lett. 107, 067002 (2011).
9. We have added derivation of the zero Kerr rotation angle in non-magnetic gyrotropic in Methods.
10. We have added a review of the bulk current response in Supplementary Note 1.
11. We have added model calculations of the band structure and the dielectric function in the form of Supplementary Figures 1 and 2.

REVIEWERS' COMMENTS

Reviewer #1 (Remarks to the Author):

The reply and revised manuscript have well answered my questions and comments. Thus, I recommend it for publication.

Reviewer #2 (Remarks to the Author):

I am grateful to the authors for their straight responses to my questions, and all the additional work they put into the manuscript.

It appears they confirm that my understanding of the significance of their work ("points 1-5" of the initial review), and that the bulk phenomena they describe are well-known effects of spatial dispersion. They also confirm (as far as I can tell) that the surface currents they consider are indeed fully determined by the bulk nonlocal responses, the way Agranovich and Yudson described this fact (Ref. [54] in the current version of the manuscript). My guess about some cancellations can be right or wrong, as it's hard to check the algebra, but what remains true is that the work is a nice review of known phenomena, both bulk and surface (one can actually even justify a new "umbrella" name this way), with applications worked out for specific optical experiments. This is definitely a useful effort, but it belongs in a more specialized journal.

Reviewer #3 (Remarks to the Author):

The authors have revised the manuscript in response to the questions and comments of the referees. I am mostly satisfied with the authors' responses. I think that a new and interesting aspect of the optical response of topological antiferromagnets has been revealed in this work, and that the manuscript is now suitable for publication in Nature Communications.

Response to Reviewer #1

Comment: The reply and revised manuscript have well answered my questions and comments. Thus, I recommend it for publication.

Response: We thank the referee for recommending our paper for publication in Nature Communications.

Response to Reviewer #2

Comment: I am grateful to the authors for their straight responses to my questions, and all the additional work they put into the manuscript.

It appears they confirm that my understanding of the significance of their work ("points 1-5" of the initial review), and that the bulk phenomena they describe are well-known effects of spatial dispersion. They also confirm (as far as I can tell) that the surface currents they consider are indeed fully determined by the bulk nonlocal responses, the way Agranovich and Yudson described this fact (Ref. [54] in the current version of the manuscript). My guess about some cancellations can be right or wrong, as it's hard to check the algebra, but what remains true is that the work is a nice review of known phenomena, both bulk and surface (one can actually even justify a new "umbrella" name this way), with applications worked out for specific optical experiments. This is definitely a useful effort, but it belongs in a more specialized journal.

Response: We thank the referee for the positive evaluation. In particular, we appreciate pointing out that our work nicely integrates our understanding of bulk and surface responses.

We want to emphasize that we present new formulas and predictions rather than re-interpret what is known. Our main result, Eq. (8), presents a new formula. This formula allows us to calculate the axion angle in periodic lattice systems despite the ill-definedness of the bare magneto-electric coupling tensors. Equation (10) is also a new formula, which extends analogous formulas developed at zero frequency to finite frequencies. Moreover, equations (17) and (18) allow us to predict significant Kerr effects in thin-film antiferromagnets such as MnBi_2Te_4 . This is a highly nontrivial result because, in a thin film, the wavelength seems too long to resolve any spatial modulation of magnetic moments, making one expect naively that only the net magnetic moment plays the role.

We believe that these results are indeed significant intellectual progress that clarifies previously vague understanding or misunderstanding. Our theoretical work sets a concrete ground for future optical studies of antiferromagnets, with high potential for applications to fundamental scientific research on magnets and technical applications to magnetic memory devices. Thus, we believe it is worth being read by the broad readership in Nature Communications.

Response to Reviewer #3

Comment: The authors have revised the manuscript in response to the questions and comments of the referees. I am mostly satisfied with the authors' responses. I think that a new and interesting aspect of the optical response of topological antiferromagnets has been revealed in this work, and that the manuscript is now suitable for publication in Nature Communications.

Response: We thank the referee for giving high evaluations of our work and recommending it for publication in Nature Communications.